# Towards Best Practices of Activation Patching in Language Models: Metrics and Methods

**Fred Zhang**
UC Berkeley
z0@berkeley.edu

**Neel Nanda**
Independent
neelnanda27@gmail.com

## Abstract

Mechanistic interpretability seeks to understand the internal mechanisms of machine learning models, where localization—identifying the important model components—is a key step. Activation patching, also known as causal tracing or interchange intervention, is a standard technique for this task (Vig et al., 2020), but the literature contains many variants with little consensus on the choice of hyperparameters or methodology. In this work, we systematically examine the impact of methodological details in activation patching, including evaluation metrics and corruption methods. In several settings of localization and circuit discovery in language models, we find that varying these hyperparameters could lead to disparate interpretability results. Backed by empirical observations, we give conceptual arguments for why certain metrics or methods may be preferred. Finally, we provide recommendations for the best practices of activation patching going forwards.

## 1 Introduction

Mechanistic interpretability (MI) aims to unravel complex machine learning models by reverse engineering their internal mechanisms down to human-understandable algorithms (Geiger et al., 2021; Olah, 2022; Wang et al., 2023). With such understanding, we can better identify and fix model errors (Vig et al., 2020; Hernandez et al., 2021; Meng et al., 2022; Hase et al., 2023), steer model outputs (Li et al., 2023b) and explain emergent behaviors (Nanda et al., 2023a; Barak et al., 2022).

A basic goal in MI is localization: identify the specific model components responsible for particular functions. Activation patching, also known as causal tracing, interchange intervention, causal mediation analysis or representation denoising, is a standard tool for localization in language models (Vig et al., 2020; Meng et al., 2022). The method attempts to pinpoint activations that causally affect on the output. Specifically, it involves 3 forward passes of the model: (1) on a clean prompt while caching the latent activations; (2) on a corrupted prompt; and (3) on the corrupted prompt but replacing the activation of a specific model component by its clean cache. For instance, the clean prompt can be "The Eiffel Tower is in" and the corrupted one with the subject replaced by "The Colosseum". If the model outputs "Paris" in step (3) but not in (2), then it suggests that the specific component being patched is important for producing the answer (Vig et al., 2020; Pearl, 2001).

This technique has been widely applied for language model interpretability. For example, Meng et al. (2022); Geva et al. (2023) seek to understand which model weights store and process factual information. Wang et al. (2023); Hanna et al. (2023); Lieberum et al. (2023) perform circuit analysis: identify the sub-network within a model's computation graph that implements a specified behavior. All these works leverage activation patching or its variants as a foundational technique.

Despite its broad applications across the literature, there is little consensus on the methodological details of activation patching. In particular, each paper tends to use its own method of generating corrupted prompts and the metric of evaluating patching effects. Concerningly, this lack of standardization leaves open the possibility that prior interpretability results may be highly sensitive to the hyperparameters they adopt. In this work, we study the impact of varying the metrics and methods in activation patching, as a step towards understanding best practices. To our knowledge, this is the first such systematic study of the technique.

Specifically, we identify three degrees of freedom in activation patching. First, we focus on the approach of generating corrupted prompts and evaluate two prominent methods from the literature:

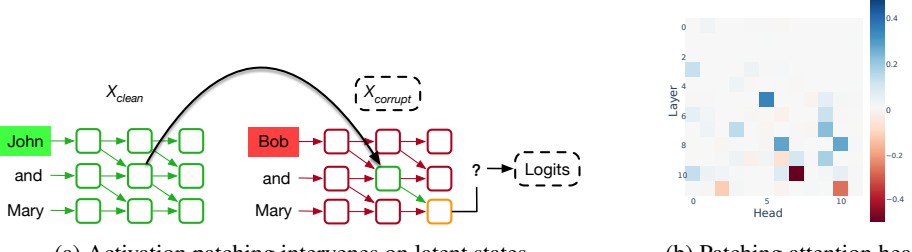

(a) Activation patching intervenes on latent states  (b) Patching attention heads

Figure 1: **The workflow of activation patching** for localization: run the intervention procedure (a) on every relevant component, such as all the attention heads, and plot the effects (b).

- Gaussian noising (GN) adds a large Gaussian noise to the token embeddings of the tokens that contain the key information to completing a prompt, such as its subject (Meng et al., 2022).
- Symmetric token replacement (STR) swaps these key tokens with semantically related ones; for example, "The Eiffel Tower"→"The Colosseum" (Vig et al., 2020; Wang et al., 2023).

Second, we examine the choice of metrics for measuring the effect of patching and compare probability and logit difference; both have found applications in the literature (Meng et al., 2022; Wang et al., 2023; Conmy et al., 2023). Third, we study sliding window patching, which jointly restores the activations of multiple MLP layers, a technique used by Meng et al. (2022); Geva et al. (2023).

We empirically examine the impact of these hyperparameters on several interpretability tasks, including factual recall (Meng et al., 2022) and circuit discovery for indirect object identification (IOI) (Wang et al., 2023), greater-than (Hanna et al., 2023), Python docstring completion (Heimersheim & Janiak, 2023) and basic arithmetic (Stolfo et al., 2023). In each setting, we apply methods distinct from the original studies and assess how different interpretability results arise from these variations.

**Findings**  Our contributions uncover nuanced discrepancies within activation patching techniques applied to language models. On corruption method, we show that GN and STR can lead to inconsistent localization and circuit discovery outcomes (Section 3.1). Towards explaining the gaps, we posit that GN breaks model's internal mechanisms by putting it off distribution. We give tentative evidence for this claim in the setting of IOI circuit discovery (Section 3.2). We believe that this is a fundamental concern in using GN corruption for activation patching. On evaluation metrics, we provide an analogous set of differences between logit difference and probability (Section 4), including an observation that probability can overlook negative model components that hurt performance.

Finally, we compare sliding window patching with patching individual layers and summing up their effects. We find the sliding window method produces more pronounced localization than single-layer patching and discuss the conceptual differences between these two approaches (Section 5).

**Recommendations for practice**  At a high-level, our findings highlight the sensitivity of activation patching to methodological details. Backed by our analysis, we make several recommendations on the application of activation patching in language model interpretability (Section 6). We advocate for STR, as it supplies in-distribution corrupted prompts that help to preserve consistent model behavior. On evaluation metric, we recommend logit difference, as we argue that it offers fine-grained control over the localization outcomes and is capable of detecting negative modules.

## 2 BACKGROUND

### 2.1 ACTIVATION PATCHING

Activation patching identifies the important model components by intervening on their latent activations. The method involves a clean prompt ($X_{\text{clean}}$, e.g.,"The Eiffel Tower is in") with an associated answer $r$ ("Paris"), a corrupted prompt ($X_{\text{corrupt}}$, e.g., "The Colosseum is in"), and three model runs:

(1) Clean run: run the model on $X_{\text{clean}}$ and cache activations of a set of given model components, such as MLP or attention heads outputs.

(2) Corrupted run: run the model on $X_{\text{corrupt}}$ and record the model outputs.

(3) Patched run: run the model on $X_{\text{corrupt}}$ with a specific model component's activation restored from the cached value of the clean run (Figure 1a).

Finally, we evaluate the patching effect, such as $\mathbb{P}(\text{"Paris"})$ in the patched run (3) compared to the corrupted run (2). Intuitively, corruption hurts model performance while patching restores it. Patching effect measures how much the patching intervention restores performance, which indicates the importance of the activation. We can iterate this procedure over a collection of components (e.g., all attention heads), resulting in a plot that highlights the important ones (Figure 1b).

**Corruption methods**   To generate $X_{\text{corrupt}}$, GN adds Gaussian noise $\mathcal{N}(0, \nu)$ to the embeddings of certain key tokens, where $\nu$ is 3 times the standard deviation of the token embeddings from the textset. STR replaces the key tokens by similar ones with equal sequence length. In STR, let $r'$ denote the answer of $X_{\text{corrupt}}$ ("Rome"). All implementations of STR in this paper yield in-distribution prompts such that $X_{\text{corrupt}}$ is identically distributed as a fresh draw of a clean prompt.

**Metrics**   The patching effect is defined as the gap of the model performance between the corrupted and patched run, under an evaluation metric. Let cl, $*$, pt be the clean, corrupted and patched run.

- Probability: $\mathbb{P}(r)$; e.g., $\mathbb{P}(\text{"Paris"})$. The patching effect is $\mathbb{P}_{\text{pt}}(r) - \mathbb{P}_*(r)$;
- Logit difference: $\text{LD}(r, r') = \text{Logit}(r) - \text{Logit}(r')$; e.g., $\text{Logit}(\text{"Paris"}) - \text{Logit}(\text{"Rome"})$.
  The patching effect is given by $\text{LD}_{\text{pt}}(r, r') - \text{LD}_*(r, r')$. Following Wang et al. (2023), we always normalize this by $\text{LD}_{\text{cl}}(r, r') - \text{LD}_*(r, r')$, so it typically lies in $[0, 1]$, where 1 corresponds to fully restored performance and 0 to the corrupted run performance.
- KL divergence: $D_{\text{KL}}(P_{\text{cl}} || P)$, the Kullback-Leibler (KL) divergence from the probability distribution of model outputs in the clean run. The patching effect is $D_{\text{KL}}(P_{\text{cl}} || P_*) - D_{\text{KL}}(P_{\text{cl}} || P_{\text{pt}})$.

GN does not provide a corrupted prompt with a well-defined answer $r'$ ("Rome"). To make a fair comparison, the same $r'$ is used for evaluating the logit difference metric under GN.

### 2.2   Problem settings

**Factual recall**   In the setting of factual association, the model is prompted to fill in factual information, e.g., "The Eiffel Tower is in". Meng et al. (2022) posits that Transformer-based language models complete factual recall (i) at middle MLP layers and (ii) specifically at the processing of the subject's last token. In this work, we do not treat the hypothesis as ground-truth but rather reevaluate it using other approaches than what was attempted by Meng et al. (2022).

**IOI**   An IOI sentence involves an initial dependent clause, e.g., "When John and Mary went to the office", followed by a main clause, e.g., "John gave a book to Mary." In this case, the indirect object (IO) is "Mary" and the subject (S) "John". The IOI task is to predict the final token in the sentence to be the IO. We use S1 and S2 to refer to the first and second occurrences of the subject (S).

We let $p_{\text{IOI}}$ denote the distribution of IOI sentences of Wang et al. (2023) containing single-token names. GPT-2 small performs well on $p_{\text{IOI}}$ and Wang et al. (2023) discovers a circuit within the model for this task. The circuit consists of attention heads. This is also the focus of our experiments, where we uncover nuanced differences when using different techniques to replicate their result.

## 3   Corruption methods

In this section, we evaluate GN and STR on localizing factual recall in GPT-2 XL and discovering the IOI circuit in GPT-2 small.

**Experiment setup**   For factual recall, we investigate Meng et al. (2022)'s hypothesis that model computation is concentrated at early-middle MLP layers (by processing the last subject token). Specifically, we corrupt the subject token(s) to generate $X_{\text{corrupt}}$. In the patched run, we override the MLP activations at the last subject token. Following Meng et al. (2022); Hase et al. (2023), at each layer we restore a set of 5 adjacent MLP layers. (More results on other window sizes can be found in Section H.1. We examine sliding window patching more closely in Section 5.)

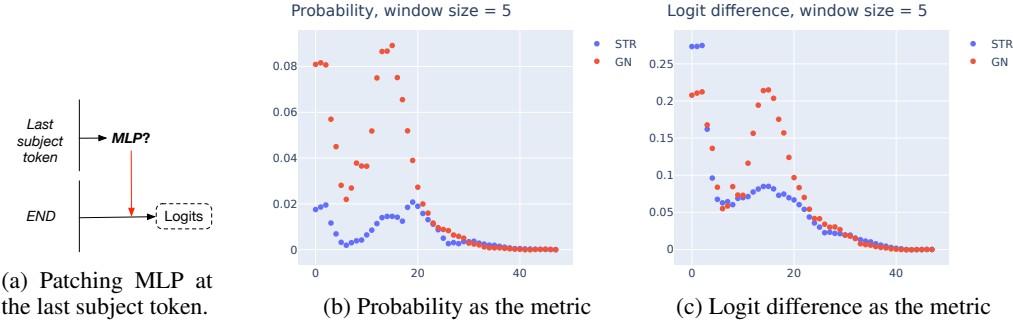

(a) Patching MLP at the last subject token.    (b) Probability as the metric    (c) Logit difference as the metric

Figure 2: **Disparate MLP patching effects for factual recall in GPT-2 XL**. (a) We patch MLP activations at the last subject token. (b)(c) The patching effects using different corruption methods with a window size of 5. STR suggests much a weaker peak, regardless of the evaluation metric.[1]

For IOI circuit discovery, we follow Wang et al. (2023) and focus on the role of attention heads. Corruption is applied to the S2 token. Then we patch a single attention head's output (at all positions) and iterate over all heads in this way. To avoid relying on visual inspection, we say that a head is *detected* if its patching effect is 2 standard deviations (SD) away from the mean effect.

**Dataset and corruption method**   STR requires pairs of $X_{\text{clean}}$ and $X_{\text{corrupt}}$ that are semantically similar. To perform STR, we construct PAIREDFACTS of 145 pairs of prompts on factual recall. All the prompts are in-distribution, as they are selected from the original dataset of Meng et al. (2022); see Appendix C for details. GPT-2 XL achieves an average of 49.0% accuracy on this dataset.

For the IOI circuit, we use the $p_{\text{IOI}}$ distribution to sample the clean prompts. For STR, we replace S2 by IO to construct $X_{\text{corrupt}}$ such that $X_{\text{corrupt}}$ is still a valid in-distribution IOI sentence. For GN, we add noise to the S2's token embedding. The experiments are averaged over 500 prompts.

### 3.1 RESULTS ON CORRUPTION METHODS

**Difference in MLP localization**   For patching MLPs in the factual association setting, Meng et al. (2022) show that the effects concentrate at early-middle layers, where they apply GN as the corruption method. Our main finding is that the picture can be largely different by switching the corruption method, regardless of the choice of metric. In Figure 2, we plot the patching effects for both metrics. Notice that the clear peak around layer 16 under GN is not salient at all under STR.

This is a robust phenomenon: across window sizes, we find the peak value of GN to be 2×–5× higher than STR; see Appendix H.1 for further plots on GPT-2 XL in this setting.

These findings illustrate potential discrepancies between the two corruption techniques in drawing interpretability conclusions. We do not, though, claim that results from GN are illusory or overly inflated. In fact, GN does not always yield sharper peaks than STR. For certain basic arithmetic tasks in GPT-J, STR can show stronger concentration in patching MLP activations; see Appendix D.

**Difference in circuit discovery**   We focus on discovering the main classes of attention heads in the IOI circuit, including (Negative) Name Mover (NM), Duplicate Token (DT), S-Inhibition (SI), and Induction Heads. The results are summarized in Table 1 and more details in Appendix I.

Most importantly, we observe that STR and GN produce inconsistent discovery results. In particular, for any fixed metric, STR and GN detect different sets of heads as important, highlighted in Table 1.

We remark that all the detections are in the IOI circuit as found by Wang et al. (2023); see Appendix B for an overview. However, the discovery we achieved here appear far from complete, with some critical misses such as NM. This suggests that the extensive manual inspection and the use of path patching, a more surgical patching method, are both necessary to fully discover the IOI circuit.

We also validate our high-level conclusions on the Python docstring (Heimersheim & Janiak, 2023) and the greater-than (Hanna et al., 2023) task. In particular, we find GN can produce highly noisy localization outcomes in these settings; see Appendix E and Appendix F for details.

---

[1]The effects on the first 3 layers are large simply because MLP0 has significant influence on the model's outputs in GPT-2, regardless of the task (Wang et al., 2023; Hase et al., 2023), so it is not the focus here.

| Corruption | Metric | NM | DT | SI | Negative NM | Induction |
|---|---|---|---|---|---|---|
| STR | Probability | **1/3** | **0/2** | **3/4** | **1/2** | 1/2 |
| GN [†] | Probability | **0/3** | **1/2** | **2/4** | **2/2** | 1/2 |
| STR | Logit difference | **1/3** | **0/2** | 3/4 | 2/2 | 1/2 |
| GN | Logit difference | 1/3 | **1/2** | 3/4 | 2/2 | 1/2 |
| STR | KL divergence | **1/3** | 0/2 | **3/4** | 2/2 | 1/2 |
| GN [†] | KL divergence | **0/3** | 0/2 | **2/4** | 2/2 | 1/2 |

Table 1: **Inconsistency on the IOI task**. We patch the attention heads outputs and list the detections of each class; e.g., 1/3 NM indicates 1 out of 3 NMs is detected. [†]Also detect 0.10, a fuzzy Duplicate Token Head, as *negatively* influencing model performance. We expect it to be positive (Wang et al., 2023).

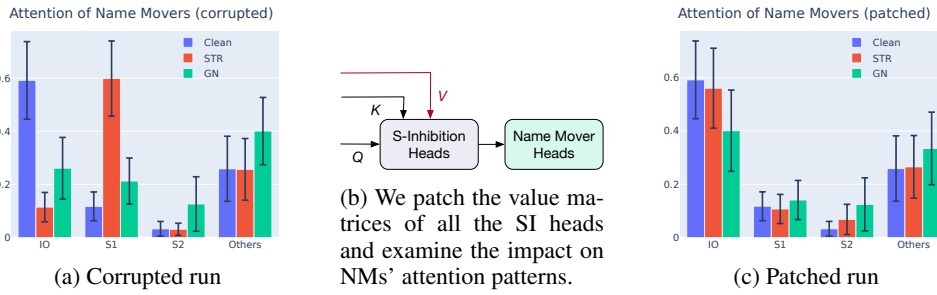

(a) Corrupted run

(b) We patch the value matrices of all the SI heads and examine the impact on NMs' attention patterns.

(c) Patched run

Figure 3: **Attention of the Name Movers** from the last token, in corrupted and patched runs.

## 3.2 Evidence for OOD behavior in Gaussian noise corruption

We suspect that the gaps between the corruption methods can be attributed partly to model's OOD behavior under GN corruption. In particular, the Gaussian noise may break model's internal mechanisms by introducing OOD inputs to the layers. We now give some tentative evidence for this hypothesis. Following the notation of Wang et al. (2023), a head is denoted by "layer.head". We say a detection is negative if the patching effect of the component is negative (under a given metric).

**Negative detection of 0.10 under GN** Although most localizations we obtain above seem aligned with the findings of Wang et al. (2023), a major anomaly in the GN experiment is the "negative" detection of head 0.10. In particular, probability and KL divergence suggest that it contributes negatively to model performance. (Logit difference also assigns a negative effect, though to a lesser degree; see Figure 29b.) This is not observed at all in the experiments with STR corruption.

The detection is in the wrong direction, given the evidence from Wang et al. (2023) that 0.10 *helps* with IOI; on clean prompts, it is active at S2, attends to S1 and signals this duplication. However, by visualizing the attention patterns, we find that this effect largely disappears under GN corruption. We intuit that the Gaussian noise is strongest at influencing early layers, and 0.10's behavior may be broken here, since it directly receives the noised token embeddings from the residual stream.

**Attention of Name Movers** To exhibit the OOD behavior of the model internals under GN corruptions, we examine the Name Mover (NM) Heads, a class of attention heads that directly affects the model's logits in the IOI circuit (Wang et al., 2023). NMs are active at the last token and copy what they attend to. We plot the attention of NMs in clean and corrupted runs in Figure 3a.

Indeed, on 500 clean IOI prompts, the NMs assign an average of 0.58 attention probability to IO. In the corrupted runs, since STR simply exchanges IO by S1, the attention patterns of NMs are preserved (with the role of IO and S1 switched). On the other hand, with GN corruption, we see that the attention is shared between IO and S1 (0.26 and 0.21). This suggests that GN not only removes the relevant information but also disrupts the internal mechanism of NMs on IOI sentences.

To take a deeper dive, Wang et al. (2023) shows that the output of NMs is determined largely by the values of the S-Inhibition Heads. Indeed, we can fully recover model's logit on IO in STR (logit difference: 1.04) by restoring the values of the S-Inhibition Heads (Figure 3b). The same intervention, however, is fairly unsuccessful under GN (logit difference: 0.49).

Towards explaining this gap, we again examine the attention of NMs. Figure 3c shows that patching nearly restores the NMs' in-distribution attention pattern under STR, but fails under GN corruption. We speculate that GN introduces further corrupted information flowing into the NMs such that restoring the clean activations of S-Inhibition Heads cannot correct their behaviors.

## 4    EVALUATION METRICS

We now study the choice of evaluation metrics in activation patching. We perform two experiments that highlight potential gaps between logit difference and probability. Along the way, we provide a conceptual argument for why probability can overlook negative components in certain settings.

### 4.1    LOCALIZING FACTUAL RECALL WITH LOGIT DIFFERENCE

The prior work of Meng et al. (2022) hypothesizes that factual association is processed at the last subject token. Motivated by this claim, we extend our previous experiments to patching the MLP outputs at all token positions and consider the effect of changing evaluation metrics.

**Experimental setup**    We apply the same setting as in Section 3. We extend our MLP patching experiments to all token positions and again use logit difference and probability as the metric.

**Experimental results**    For STR and window size of 5, we plot the patching effects across layers and positions in Figure 4. The visualization shows that probability assigns stronger effects at the last subject token than logit difference. Specifically, we calculate the ratio between the sum of effects (over all layers) on the last subject token and those on the middle subject tokens. In both corruptions, probability assigns more effects to the last subject token than logit difference:

- Using STR corruption, the ratio is 4.33× in probability > 1.22× in logit difference.
- Using GN corruption, the ratio is 1.74× in probability > 0.77× in logit difference.

This observation holds for other window sizes, too, for which we provide details in Appendix H.2. We also validate our findings on GPT-J 6B (Wang & Komatsuzaki, 2021) in Appendix H.5. The results show that the choice of evaluation metrics influences the patching effects across tokens.

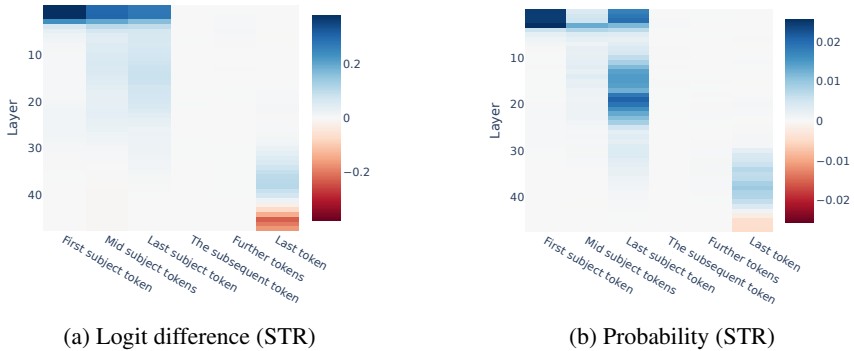

(a) Logit difference (STR)                    (b) Probability (STR)

Figure 4: **Activation patching on MLP** across layers and token positions in GPT-2 XL, with a sliding window patching of size 5. Note that probability (b) highlights the importance of the last subject token, whereas logit difference (a) displays less effects.

### 4.2    CIRCUIT DISCOVERY WITH PROBABILITY

Wang et al. (2023) discovers two Negative Name Mover (NNM) heads, 10.7 and 11.10, that noticeably hurt model performance on IOI. In our previous experiments on STR, both are detected, except when using probability as the metric where 11.10 is overlooked. In fact, the patching effect of 11.10 under STR in probability is well within 2 SD from the mean (mean 0.003, SD 0.015, and 11.10 receives −0.022). Looking closely, the reason is simple:

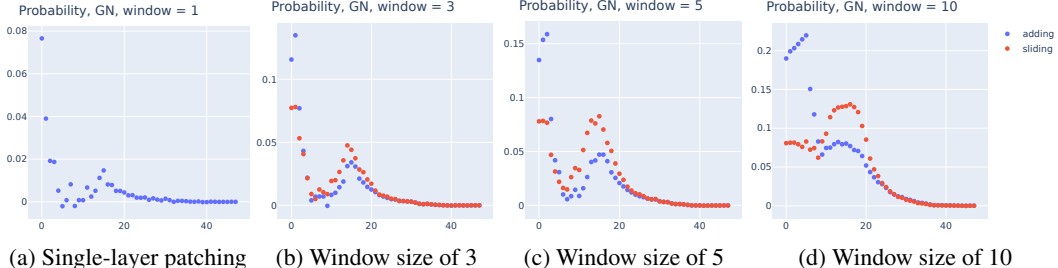

(a) Single-layer patching    (b) Window size of 3    (c) Window size of 5    (d) Window size of 10

Figure 5: **Sliding window patching vs summing up individual patching effects**; patching MLP activation at the last subject token in GPT-2 XL on factual recall prompts. Sliding window patching offers $1.40\times$, $1.75\times$ and $1.59\times$ peak value than summation of single-layer patchings. Single-layer patching (a) suggests a weak peak.

- In the corrupted run of STR, the average probability of outputting the original IO is $0.03$. Hence, the patching effect in probability, $\mathbb{P}_{pt}(IO) - \mathbb{P}_*(IO)$, is at least $-0.03$, as $\mathbb{P}_{pt}(IO)$ is non-negative. This is already close to 2 SD below the mean ($-0.027$). Hence, for an NNM to be detected via patching, its $\mathbb{P}_{pt}(IO)$ needs to be near $0$, which may be hard to reach.
- By contrast, under GN corruption, the average probability of IO is $0.13$. Intuitively, this makes a lot more space for NNMs to demonstrate their effects.

In general, probability must fail to detect negative model components, if corruption reduces the correct token probability to near zero. We now give a cleaner experimental demonstration of this concern, using an original approach of Wang et al. (2023).

**Experimental setup**  We revisit an alternative corruption method proposed by Wang et al. (2023), where S1, S2 and IO are replaced by three unrelated random names[2]; for example, "John and Mary [...], John" → "Alice and Bob [...], Carol." We use probability of the original IO as the metric. Intuitively, this replacement method would achieve much stronger corruption effect, since it removes all the relevant information (S and IO) of the original IOI sentence.

**Experimental results**  First, we observe that the probability of outputting the IO of the original IOI sentence is negligible ($5e{-}4$) under this corruption. As a result, using probability detects neither NNMs. On the other hand, we find that logit difference still can. See Appendix I.3 for the plots. In Appendix G, we confirm the same finding when corruption is applied to S1 and IO only.

At a high-level, we believe that this is a pitfall of probability as an evaluation metric. Its non-negative nature makes it incapable of discovering negative model components in certain settings.

## 5   SLIDING WINDOW PATCHING

In this section, we examine the technique of sliding window patching in localizing factual information (Meng et al., 2022). For each layer, the method patches multiple adjacent layers simultaneously and computes the joint effects. Hence, one should interpret the result of Meng et al. (2022) as the effects being constrained within a window rather than at a single layer. We argue that such as hypothesis can be tested by an alternative approach and we compare the results from these two.

**Experimental setup**  Instead of restoring multiple layers simultaneously, we patch each individual MLP layer one at a time. Then as an aggregation step, for each layer, sum up the single-layer patching effects of its adjacent layers. For example, we add up the effect at layer 2 to layer 6 to get an aggregated effect for layer $4$. We patch the MLP output at the last subject token.

**Experimental results**  For each window size, we compute the ratio of the maximum patching effect at the middle MLP layers between sliding window patching and summation of single-layer patching. Over the combinations of window sizes, metrics and corruption methods, we find sliding window patching typically provides at least $20\%$ more peak effect than the summation method.

---

[2]This corrupted distribution is denoted by $p_{ABC}$ in the original paper of Wang et al. (2023)

In Figure 5, for window sizes of $3, 5, 10$, we plot the results using GN corruption and probability as the metric, the original setting as in Meng et al. (2022). We observe significant gaps between the sliding window and the summation method. Moreover, for single-layer patching, the peak at layer 15 is fairly weak (Figure 5a). Sliding window patching appears to generate more pronounced the concentration, as we increase the window sizes.

The result suggests that sliding window patching tends to amplify weak localization from single-layer patching (see Figure 12 for plots on single-layer MLP patching in GPT-2 XL). We believe this may arise due to certain non-linear effects in joint patching and therefore results from which should be carefully interpreted; see Section 6 for more discussions.

## 6 DISCUSSION AND RECOMMENDATIONS

We have observed a variety of gaps between corruption methods and evaluation metrics used in activation patching on language models. In this section, we summarize our findings and provide recommendations.

**Corruption methods**  We are concerned that GN corruption puts the model off distribution by introducing noise never seen during training. Indeed, in Section 3.2, we provide evidence that in the corrupted run, model's internal functioning is OOD relative to the clean distribution. This may induce unexpected anomalies in the model behavior, interfering with our ability to localize behavior to specific components. Conceivably, GN corruption could even lead to unreliable or illusory results.

More broadly, this presents a challenge to any intervention techniques that introduce OOD inputs to the model or its internal layers, including ablations. In fact, similar concerns have been raised earlier in the interpretability literature on feature attribution as well; see e.g. Hooker et al. (2019); Janzing et al. (2020); Hase et al. (2021).

In contrast, STR uses counterfactual prompts ("The Eiffel Tower is in" vs "The Colosseum is in") that are in-distribution and thus induces in-distribution activations, avoiding the OOD issue. Thus, we recommend STR whenever possible. GN, or simpler methods such as ablation, may be considered as an alternative when token alignment or lack of analogous tokens makes STR unsuitable.

**Evaluation metrics**  We generally recommend avoiding using probability as the metric, given that it may fail to detect negative model components.

We find logit difference a convincing metric for localization in language models. Consider an IOI setting where a model contains an attention head that boosts the logits of all (single-token) names. This head, though important, should not be viewed as part of the IOI circuit, but our interventions may still affect it.[3] By measuring $\text{Logit}(\text{IO}) - \text{Logit}(\text{S})$, logit difference controls for such components and ensures they are not detected. This may not be achieved by other metrics, such as probability or $\text{Logit}(\text{IO})$ alone.

KL divergence tracks the full model output distributions, rather than focused only on the correct or incorrect answer, and can be a reasonable metric for circuit discovery as well (Conmy et al., 2023).

**Sliding window patching**  We speculate that simultaneously patching multiple layers could capture the following non-linear effects and results in inflated localization plots:

- Joint patching may suppress the flow of corrupted information within the window of patched layers, where single-layer patching offers no such control.
- A window of patched layers may jointly perform a crucial piece of computation, such as a major boost to the logit of the correct token, which no individual layer can single-handedly achieve.

Generally, when examining the outcome from sliding window patching, one should be aware of the possibility of multiple layers working together. Thus, the results from the technique are to be interpreted as the joint effects of the full window, rather than of a single layer. In practice, we recommend experimenting with single-layer patching first and only consider sliding window patching when individual layers seem to induce small effects.

---

[3]We note that if our interventions do not affect the head, then it will not show up on any metric.

**Which tokens to corrupt?**   In some problem settings, a prompt contains multiple key tokens, all relevant to completing the task. This would offer the flexibility to choose which tokens to corrupt. This is another important dimension of activation patching. For instance, our experiments on IOI in Section 3 corrupt the S2 token. An alternative is to corrupt the S1 and IO. While this may seem an implementation detail, we find that this can greatly affect the localization outcomes.

Specifically, in Appendix G, we test corrupting S1 and IO in activation patching on IOI sentences, by changing their values to random names or adding noise to the token embeddings . We find that almost all techniques discover the 3 Name Mover (NM) Heads of the IOI circuit (Table 4 and Figure 11). These are attention heads that directly contribute to Logit(IO) as shown by Wang et al. (2023). In contrast, our prior experiments corrupting S2 miss most of them (Table 1).

We intuit that corrupting different tokens allows activation patching to trace different information within the model, thereby suggesting varying localizations results. For instance, in our prior experiments replacing S2 by IO, patching traces the value of IO or its position. On the other hand, in changing the values of S1 and IO while fixing their positions, patching highlights exactly where the model processes these values.

In practice, we recommend trying out different tokens to corrupt when the problem setting offers such flexibility. This may lead to more exhaustive circuit discovery.

## 7   RELATED WORK

**Activation patching**   Activation patching is a variant of causal mediation analysis (Vig et al., 2020; Pearl, 2001), similar forms of which are used broadly in the interpretability literature (Soulos et al., 2020; Geiger et al., 2020; Finlayson et al., 2021; Geiger et al., 2022). The specific one with GN corruption was first proposed by Meng et al. (2022) under the name of causal tracing. Wang et al. (2023); Goldowsky-Dill et al. (2023) generalize this to a more sophisticated version of path patching.

**Circuit analysis**   Circuit analysis provides post-hoc model interpretability (Casper et al., 2022). This line of work is inspired by Cammarata et al. (2020); Elhage et al. (2021). Other works include Geva et al. (2022); Li et al. (2023a); Nanda et al. (2023a); Chughtai et al. (2023); Zhong et al. (2023); Nanda et al. (2023b); Varma et al. (2023); Wen et al. (2023); Hanna et al. (2023); Lieberum et al. (2023). Circuit analysis often requires manual effort by researchers, motivating recent work to scale or automate parts of the workflow (Chan et al., 2022; Bills et al., 2023; Conmy et al., 2023; Geiger et al., 2023; Wu et al., 2023; Lepori et al., 2023).

**Mechanistic interpretability (MI)**   MI aims to explain the internal computations and representations of a model. While circuit analysis is a major direction under this broad theme, other recent case studies of MI in language model include Mu & Andreas (2020); Geva et al. (2021); Yun et al. (2021); Olsson et al. (2022); Scherlis et al. (2022); Dai et al. (2022); Gurnee et al. (2023); Merullo et al. (2023); McGrath et al. (2023); Bansal et al. (2023); Dar et al. (2023); Li et al. (2023c); Brown et al. (2023); Katz & Belinkov (2023); Cunningham et al. (2023).

## 8   CONCLUSION

We examine the role of metrics and methods in activation patching in language models. We find that variations in these techniques could lead to different interpretability results. We provide several recommendations towards the best practice, including the use of STR as the corruption method.

In terms of limitations, our experiments are on decoder-only language models of size up to 6B. We leave it as a future direction to study other architectures and even larger models. Our work tests overriding corrupted activations by clean activations. The other direction—patching corrupted to clean—has also been used for circuit discovery, and it is interesting to compare these two. In addition, we provide tentative evidence that certain corruption methods lead to OOD model behaviors and suspect that this can make the resulting interpretability claims unreliable. Future work should examine this hypothesis closely and furnish further demonstrations. Finally, it is interesting to develop more principled techniques for activation patching or propose other methods for localization.

ACKNOWLEDGMENTS

FZ would like to thank Matthew Farhbach, Dan Friedman, Johannes Gasteiger, Asma Ghandehari-oun, Stefan Heimersheim, János Kramár, Kaifeng Lyu, Vahab Mirrokni, Jacob Steinhardt and Peilin Zhong for helpful discussions, and Jiahai Feng, Yossi Gandelsman, Oscar Li and Alex Wei for comments on early drafts of the paper.

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

## A  REVIEW OF THE TRANSFORMER ARCHITECTURE

We follow the notation of Elhage et al. (2021) and give a review of the Transformer architecture (Vaswani et al., 2017). The input $x_0 \in \mathbb{R}^{N \times d}$ to a transformer model is a sum of position and token embeddings, where $N$ is the sequence length and $d$ is the dimension of the model's internal states. The input is the initial value of the residual stream which subsequently gets updated by the transformer blocks.

Each transformer block consists of a multi-head self-attention sublayer and an MLP sublayer. (For GPT-J, these two sublayers are parallelized.) The MLP sublayer is a two-layer feedforward network that processes each token position independently in parallel. Following Elhage et al. (2021), the output of the attention sublayer can be decomposed into individual heads. For the $i$th layer, the attention output can be written as $y_i = \sum_{j=1}^{H} h_{i,j}(x_i)$, where $h_{i,j}$ denotes the $j$th attention head of the layer. Each head has four weight matrices, $W_Q^{i,j}, W_K^{i,j}, W_V^{i,j} \in \mathbb{R}^{d \times \frac{d}{H}}$ and $W_O \in \mathbb{R}^{\frac{d}{H} \times d}$. For

a residual stream $x$, we refer to $Q^{i,j} = xW_Q^{i,j}, K^{i,j} = xW_K^{i,j}, V^{i,j} = xW_V^{i,j}$ as the query, key and value of the head. The attention pattern is given by

$$A^{i,j} = \text{softmax}\left(\frac{(xW_Q^{i,j})(xW_K^{i,j})^T}{\sqrt{d/H}} + M\right) \in \mathbb{R}^{N \times N},$$

where $M$ is the attention mask. In auto-regressive language models, the attention pattern is masked to a lower triangular matrix. The output of the attention sublayer is given by

$$x + \text{Concat}\left[A^{i,1}V^{i,1}, \ldots, A^{i,j}V^{i,j}, \ldots, A^{i,H}V^{i,H}\right]W_O. \tag{1}$$

## B   REVIEW OF THE ATTENTION HEADS IN IOI CIRCUIT

We provide an overview of the attention heads and their functionalities of the IOI circuit in GPT-2 small, as found in Wang et al. (2023). Our exposition here follows Section 3 of Wang et al. (2023).

- Name Mover (NM) Heads are active at the END position, attend to previous names, and copy the names they attend to. On IOI sentences, they directly copy the correct name and thus contribute positively to model performance.
- Negative Name Mover Heads writes in the opposite direction of the Name Mover Heads, decreasing the model's logit on the correct name.
- Duplicate Token (DT) Heads identify tokens that previously appeared in the sentence. On IOI sentences, they are active at S2 and attend primarily to S1, contributing positively to model performance.
- Induction Heads play the same role as the Duplicate Token Heads, though via a different induction-like mechanism.
- S-Inhibition (SI) Heads remove duplicate tokens from Name Mover Heads' attention. On IOI sentences, they inhibit Name Mover Heads' attention to S1 and S2.

Aside from Negative Name Mover Heads, all the model components above contribute positively to the model performance on $p_{\text{IOI}}$, as shown in Wang et al. (2023).

**Head 0.10**   Head 0.10 is found to be a *fuzzy* Duplicate Token Head by Wang et al. (2023). It pays attention to S1 from S2, but the pattern is fuzzy, as other tokens also receive non-negligible attention mass. Nonetheless, it is expected that it helps with the model on the IOI task. We give a detailed study on the effect of patching head 0.10 in Section 3.2 under GN corruption.

## C   DETAILS ON EXPERIMENTAL SETTINGS

For Gaussian noise (GN) corruption, we corrupt the embeddings of the crucial tokens by adding a Gaussian noise $\varepsilon \sim \mathcal{N}(0; \nu)$, where $\nu$ is set to be 3 times the standard deviation of the token embeddings from the dataset (Meng et al., 2022).

We always perform GN and STR experiments in parallel. For STR, there is a natural the incorrect token $r'$, since $X_{\text{corrupt}}$ is also a valid in-distribution prompt. This allows for a well-defined metric of logit difference $\text{LD}(r, r') = \text{Logit}(r) - \text{Logit}(r')$. To make a fair comparison, the same $r'$ is used for evaluating the logit difference metric under GN.

Throughout the paper, layers are zero-indexed, numbered from $0$ to $L-1$ rather than $1$ to $L$.

**Factual recall**   To perform STR in the factual association setting, we construct PAIREDFACTS, a dataset of 145 pairs of prompts. Within each pair, the two prompts have the same sequence length (under the GPT-2 tokenizer) but distinct answers. All the prompts are selected from the COUNTERFACT and KNOWN1000 datasets of Meng et al. (2022). On these prompts,

- GPT-2 XL achieves an average of $49.0\%$ probability on the correct token and $6.85$ logit difference.
- GPT-2 large achieves $41.1\%$ and $5.88$ logit difference.

- GPT-J achieves $50.1\%$ and $7.36$ logit difference.

A few samples of the PAIREDFACTS dataset are listed in Figure 31 of Appendix J.

Since the prompts are perfectly symmetric and all of them are in-distribution, our STR experiments consist of both ways, where a prompt within a pair play the role of both $X_{\text{corrupt}}$ and $X_{\text{clean}}$.

Our experiments with GN corruption is performed in the same manner as in Meng et al. (2022); Hase et al. (2023), with noise applied to all subject tokens' embeddings.

The experiments here are implemented via the TransformerLens library (Nanda & Bloom, 2022).

**IOI**   Unless specified otherwise, GN applies Gaussian noise to the S2 token embedding. Over 500 prompts, the probability of outputting IO is $\mathbb{P}_*(r) = 0.129$ under GN corruption (with $r$ being IO), whereas it is $0.481$ under the clean distribution $p_{\text{IOI}}$.

All our experiments are performed using the original codebase of Wang et al. (2023), available at `https://github.com/redwoodresearch/Easy-Transformer`. The code provides the functionality of constructing $X_{\text{corrupt}}$ under various definitions of corruptions, including STR.

## D   RESULTS ON ARITHMETIC REASONING IN GPT-J

**Experimental setup**   We follow the setting of Stolfo et al. (2023) and perform localization analysis on the task of basic arithmetic in GPT-J (Wang & Komatsuzaki, 2021), a decoder-only model with 6B parameters. For simplicity, we consider addition, subtraction and multiplication up to 3 digits. We provide the model with a 2-shot prompts of the format

$$X_1 + Y_1 = Z_1$$
$$X_2 + Y_2 = Z_3$$
$$X_3 + Y_3 =$$

where the numbers $X_i, Y_i$ are random integers and the operator can be $+, -, \times$. Stolfo et al. (2023) finds that this leads to improved accuracy. Since large integers get split into multiple tokens, we draw $X_i, Y_i$ from $\{1, 2, \cdots, 250\}$ for addition and subtraction and from $\{1, 2, \cdots, 23\}$ for multiplication. To obtain a dataset for activation patching, we first draw 200 prompts and discard those on which the model's top-ranked output token is incorrect.

We set GN corruption to add noise to the token embeddings at the positions of $X_3, Y_3$. Similarly, STR replaces $X_3, Y_3$ by two random integers drawn from the same set, which ensures that the corrupted prompt is still in-distribution. We remark that Stolfo et al. (2023) applies the same STR corruption in their patching experiments.

Stolfo et al. (2023) devises a new metric to evaluate the patching effects. More precisely, they report:

$$\frac{1}{2}\left[\frac{\mathbb{P}_{\text{pt}}(r) - \mathbb{P}_*(r)}{\mathbb{P}_*(r)} + \frac{\mathbb{P}_*(r') - \mathbb{P}_{\text{pt}}(r')}{\mathbb{P}_{\text{pt}}(r')}\right] \tag{2}$$

from patching the MLP activation at last token of the prompt.[4] We compute the patching effect given by the metric, as well as probability and logit difference.

Following Stolfo et al. (2023), we narrow our focus on localization of MLP layers. All the experiments patch a single MLP layer's activation at the last token of the prompt.

**Experimental results**   Focused on the logit difference and probability metric, we observe gaps between GN and STR for addition and subtraction. In particular, STR is found to provide sharper concentration, up to a magnitude of $4\times$. This in contrast with our results on factual association (Section 3.1), where GN appears to induce stronger peak. For multiplication, GN and STR provides nearly matching results. This highlights that activation patching can be sensitive to corruption methods in a rather unpredictable way. See Figure 6, Figure 7 and Figure 8 for plots.

For the metric (2) of Stolfo et al. (2023), we qualitatively replicate their results, similar to Figure 2 of their paper, and find extremely pronounced peak with STR corruption. Towards understanding this

---

[4]Note that our notations here are different from Stolfo et al. (2023).

observation, we examine the quantity (2) closely and discover that its first term typically dominates the second. This, in turn, is because the denominator term $\mathbb{P}_*(r)$, the probability of outputting the correct answer in the corrupted run, is usually tiny under STR corruption. The small denominator, therefore, acts as a large multiplier that amplifies the absolute gap between patching different layers. We note that this effect is much smaller under GN since $\mathbb{P}_*(r)$ is usually not negligible.

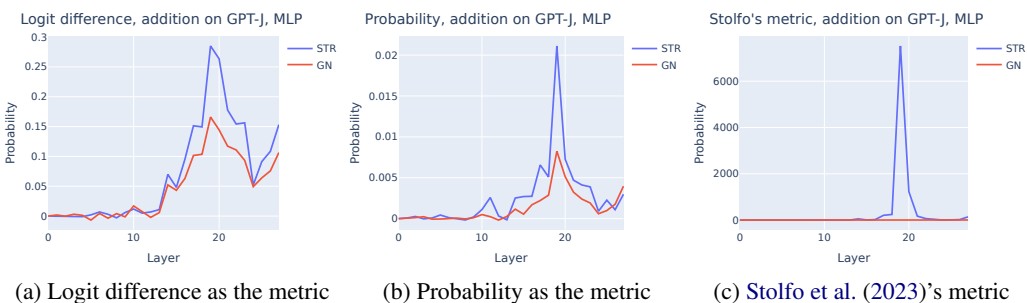

(a) Logit difference as the metric    (b) Probability as the metric    (c) Stolfo et al. (2023)'s metric

Figure 6: **The effects of patching MLP layers** in GPT-J on addition prompts.

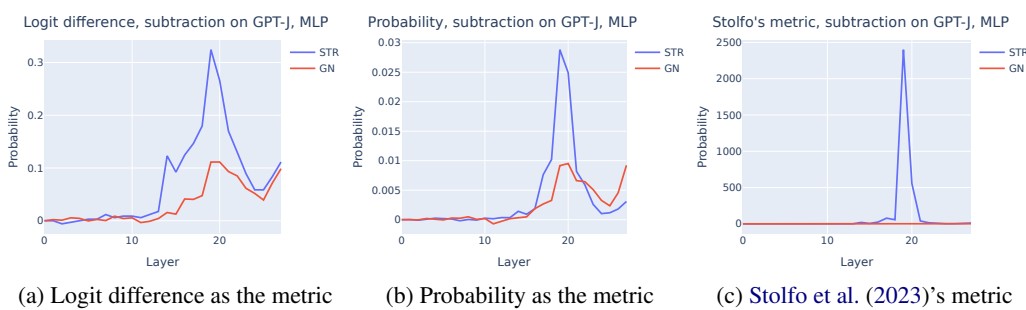

(a) Logit difference as the metric    (b) Probability as the metric    (c) Stolfo et al. (2023)'s metric

Figure 7: **The effects of patching MLP layers** in GPT-J on subtraction prompts.

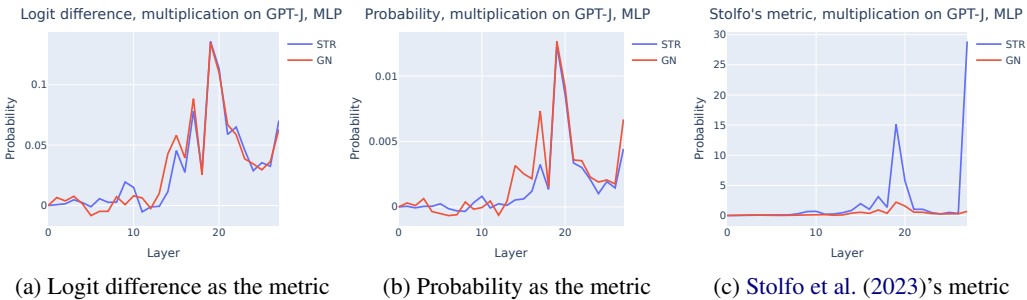

(a) Logit difference as the metric    (b) Probability as the metric    (c) Stolfo et al. (2023)'s metric

Figure 8: **The effects of patching MLP layers** in GPT-J on multiplication prompts.

## E   Results on Python docstring circuit

Heimersheim & Janiak (2023) studies a circuit for Python docstring completion in a pre-trained 4-layer attention-only Transformer.[5] We do not aim to fully replicate their results. Rather, we perform patching experiments to localize the important attention heads for the purpose of evaluating variants of activation patching.

**Experimental setup**   In Heimersheim & Janiak (2023), a Python docstring completion instance consists of the following prompt:

---

[5]The model is available in the TransformerLens library under the name `attn-only-4l` (Nanda & Bloom, 2022).

```
def rand0(self, rand1, rand2, A_def, B_def, C_def, rand3):
    """rand4 rand5 rand6

    :param A_def: rand7 rand8
    :param B_def: rand9 rand10
    :param
```

where `rand`'s are random single-token English words and the goal is to complete the prompt with `C_def`. Heimersheim & Janiak (2023) finds that the 4-layer model solves the docstring task with an accuracy of 56% and the logit difference is 0.5.

Following their approach, we run activation patching on all attention heads, across all token positions. This is more fine-grained than what we did for the IOI circuit, since the outcome would also highlight the token positions that matter for the important heads.

We apply corruption to the `C_def` token. For STR, it is replaced randomly by a single-token English word in the same way specified in Heimersheim & Janiak (2023).

**Experimental results**   We take 200 instances of the docstring completion task, perform activation patching by positions and compute the patching effects. We report all position-head pairs with patching effect 2 standard deviations away from the mean. We find that the detections are mostly at the position of `C_def` and the last token of the prompt. The details are given in Table 2.

| Corruption | Metric | At the position of C_def | At the last position |
|---|---|:---:|:---:|
| STR | Logit difference | 0.0, 0.1, 0.5 | 2.3, 3.0, 3.5, 3.6 |
| STR | Probability | | 3.0, 3.6 |
| STR | KL divergence | 0.0, 0.1, 0.5, 2.2 | 2.3, 3.0, 3.6 |
| GN † | Logit difference | 0.5 | 1.4, 1.5, 2.2, 2.3, 3.0, 3.6, 3.7 |
| GN | Probability | | 3.0, 3.6 |
| GN | KL divergence | | 2.2, 2.3, 3.0, 3.5, 3.6 |

Table 2: **Detections from activation patching** of attention heads by position on the Python docstring completion task. † Also detects two early-layer heads active at other positions and four negative heads active at the last position, which we omit here.

We again find that the localization outcomes are sensitive to the choice of corruption method and evaluation metric. The results of GN appear quite noisy, except when using probability as the metric. On the other hand, we remark that 3.0 and 3.6 are consistently highlighted across metrics and methods. In fact, they are typically assigned the largest patching effects (at the last position). This appears consistent with the result of Heimersheim & Janiak (2023), where 3.0 and 3.6 are found to be directly responsible for moving the `C_def` token.

## F   RESULTS ON THE GREATER-THAN CIRCUIT IN GPT-2 SMALL

Hanna et al. (2023) In this section, we study the greater-tan task, specified below, and perform activation patching on the attention heads in GPT-2 small. In this setting, the prior work by Hanna et al. (2023); Conmy et al. (2023) show that model computation is fairly localized in this setting and provide a set of circuit discovery results. We remark that we do not attempt to replicate the circuit discovery results here, but rather to evaluate whether activation helps with localizing certain important model components.

**Experimental setup**   Following Hanna et al. (2023), an instance of the greater-than task consists of an incomplete sentence of the template: "The <noun> lasted from the year XXYY to the year XX", where <noun> is a single-token word and XX and YY are two-digit numbers. For example, "The war lasted from year 1745 to 17". The goal is to complete the prompt with an integer greater than XX (in this case, 45). Across several metrics, Hanna et al. (2023) shows that GPT-2 small performs well on this task.

We focus on the role of attention heads in our study. To perform corruption, we ensure that the year XXYY are tokenized as [XX][YY] by filtering out years and numbers that do not conform to the constraint. GN corruption adds noise to the token embedding of YY. Following Hanna et al. (2023); Conmy et al. (2023), STR corruption replaces YY by 01. The probability metric, in this setting, is defined as the sum of probabilities of the years greater than YY. The logit difference metric is defined as the sum of logits of the years greater than YY minus the sum of logits of the years less than YY.

We perform activation patching on the attention heads outputs over all token positions.

**Experimental results** We find significant difference between the results achieved by GN and STR. In fact, the set of heads that are localized by the methods are mostly disjoint. Specifically, GN appears to give extremely noisy results that are not in line with the findings of Hanna et al. (2023); Conmy et al. (2023). The details are given in Table 3

| Corruption | Metric | Positive | Negative |
|---|---|---|---|
| STR | Logit difference | 6.9, 7.10, 8.11, 9.1, 10.4 | |
| STR | Probability | 7.10, 8.11, 9.1 | |
| STR | KL divergence | 6.9, 7.10, 8.11, 9.1 | 10.7 |
| GN | Logit difference | 0.9, 7.10, 8.10, 9.1, 10.4 | 6.1, 8.6, 9.5 |
| GN | Probability | 5.5, 6.1, 6.9, 7.10, 7.11, 8.8, 9.1 | |
| GN | KL divergence | 5.5, 6.1, 6.9, 7.10, 7.11, 8.8, 9.1 | 5.9, 7.6 |

Table 3: **Detections from activation patching on attention heads for the greater-than task** in GPT-2 small, averaged across 300 prompts.

The results from STR are fairly reasonable as the heads 6.9, 7.10, 8.11, 9.1 are also discovered by Hanna et al. (2023); Conmy et al. (2023), using more sophisticated methods. In contrast, the heads discovered by GN corruption share little overlap with STR, except 7.10 and 9.1. From visualizations, we also see that the plots for GN experiments are fairly noisy and do not yield much localization at all (Figure 9). On the other hand, the plots from STR are easily interpretable (Figure 10).

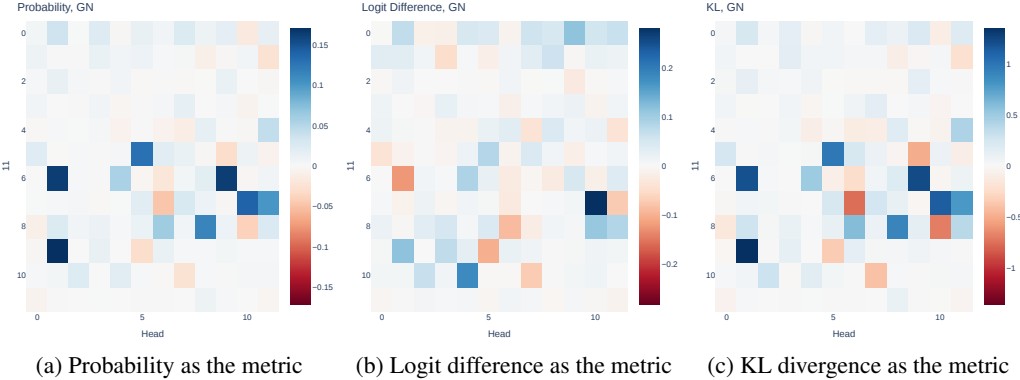

(a) Probability as the metric    (b) Logit difference as the metric    (c) KL divergence as the metric

Figure 9: **The effects of patching attention heads** in GPT-2 small on the greater-than task, using GN corruption. We see that the results are fairly noisy and do not appear to be localized.

## G   WHICH TOKENS TO CORRUPT MATTERS

In this section, we revisit the implementation of corruption methods in the setting of IOI (Wang et al., 2023).

Previously in our STR experiments in Section 3 and Section 4, the S2 token was corrupted by exchanging with IO. Similarly, in GN, we add noise to the token embedding of IO. We notice that

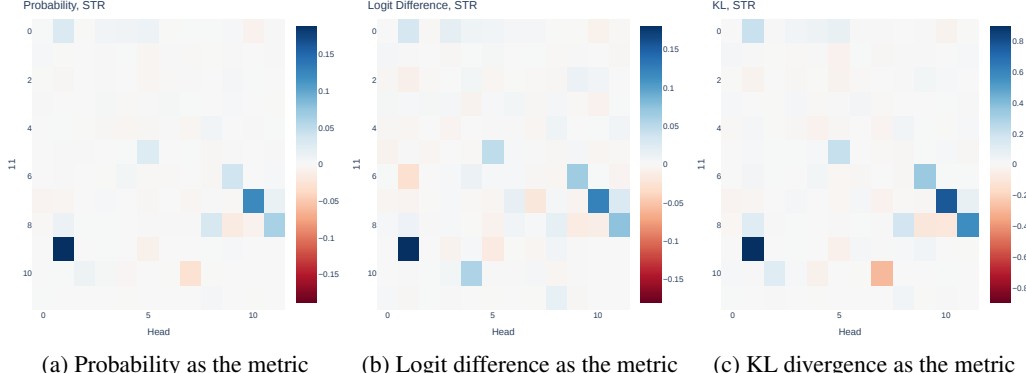

(a) Probability as the metric    (b) Logit difference as the metric    (c) KL divergence as the metric

Figure 10: **The effects of patching attention heads** in GPT-2 small on the greater-than task, using STR corruption. This gives clearly localized results.

the localization results from this approach miss at least 2 out of the 3 Name Mover (NM) Heads (Table 1); they directly contribute to the logit of IO as found by Wang et al. (2023). In particular, all combinations of metric and method would miss out on 9.6 and 10.0 (Table 5).

We show that by varying exactly which tokens to corrupt, the NMs can be discovered, too

**Experimental setup**    We consider the IOI setting (Wang et al., 2023) using STR and GN corruption for localizing attention heads. Here, we corrupt the S1 and IO tokens. For STR, we simply replace S1 and IO by two random unrelated names. In both STR and GN experiments, the S2 token remains the same as in $X_{\text{clean}}$.

We perform activation patching across all attention heads. We apply logit difference, probability and KL divergence as the metric. All the results are averaged across 500 sampled IOI sentences.

**Experimental results**    We find that most combinations of metrics and methods are able to notice all the NMs, when corruption applies to S1 and IO. We give the exact detections below and categorize them into positive and negative for simplicity.

| Corruption | Metric | Positive | Negative |
|---|---|---|---|
| STR | Logit difference | 9.6, 9.9, 10.0 | 10.7, 11.10 |
| STR | Probability | 9.9 | |
| STR | KL divergence | 9.6, 9.9, 10.0 | 10.7, 11.10 |
| GN | Logit difference | 9.6, 9.9, 10.0 | 10.7, 11.10 |
| GN | Probability | 9.6, 9.9, 10.0 | |
| GN | KL divergence | 9.6, 9.9, 10.0 | 10.7, 11.10 |

Table 4: **Detections from activation patching by corrupting S1 and IO** in IOI. The Name Mover Heads are 9.6, 9.9, 10.0 and the Negative Name Mover Heads are 10.7 and 11.10, based on Wang et al. (2023). No other heads, including the S-Inhibition Heads, are noticed with this approach.

First, we observe that this corruption seems to precisely target the NMs and their Negative counterparts. Intuitively, this is natural. Wang et al. (2023) finds that NMs write in the direction of the logit of the name (IO or S), whereas the Negative NMs do the opposite. Patching the clean activations of NMs recover such behavior.

Second, we confirm our finding that probability will miss out on the Negative NM; see Figure 11 for the plots.

Overall, the experiment suggests that exactly which token is corrupted affects the localization outcomes. Intuitively, varying the corrupted token(s) allows activation patching to trace different information within the model's computation paths; see Section 6 for a discussion.

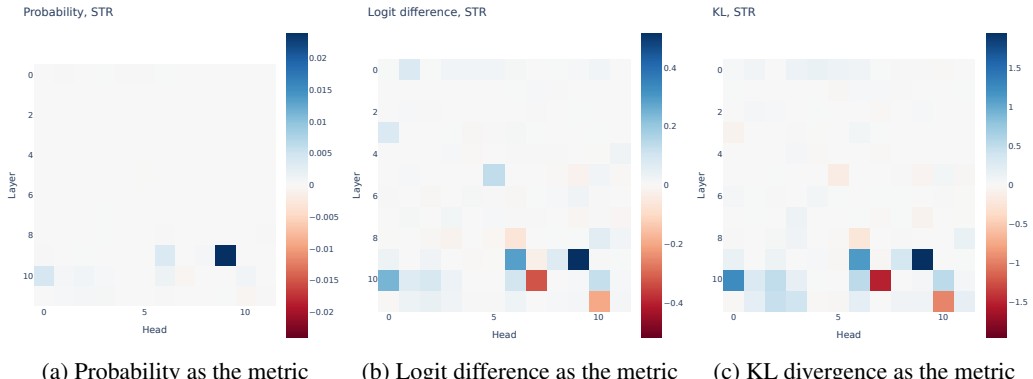

(a) Probability as the metric     (b) Logit difference as the metric     (c) KL divergence as the metric

Figure 11: **The effects of patching attention heads** in GPT-2 small on IOI sentences, using STR corruption on S1 and IO.

## H FURTHER DETAILS ON FACTUAL ASSOCIATION

The plots of subsection Appendix H.1 to H.3 are produced on GPT-2 XL and with the PARIEDFACTS as dataset. Following that, we also experiment with the GPT-2 large (Radford et al., 2019) and GPT-J (Wang & Komatsuzaki, 2021) model in Appendix H.4 and H.5.

### H.1 PLOTS ON MLP PATCHING AT THE LAST SUBJECT TOKEN IN GPT-2 XL

First, we perform single-layer patching of MLP activation at the last subject token and examine the effects in Figure 12. We observe that the experiment suggests weak or no peak at middle MLP layers, across metrics and corruption methods.

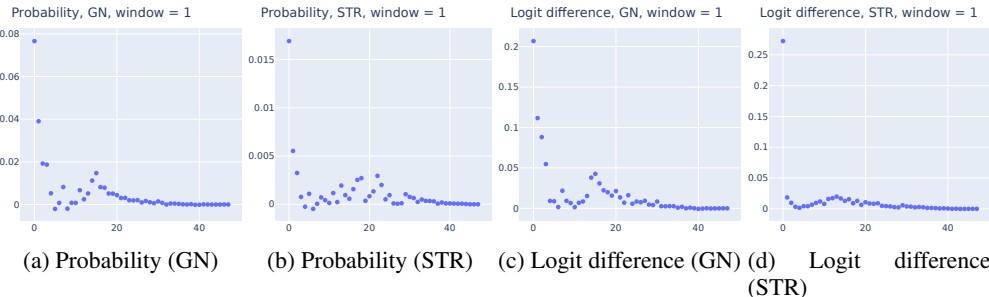

(a) Probability (GN)     (b) Probability (STR)     (c) Logit difference (GN)     (d) Logit difference (STR)

Figure 12: **Patching single MLP layers at the last subject token** in GPT-2 XL on factual recall prompts. None of them suggest a strong peak at the middle MLP layers.

Also, see Figure 13 and Figure 14 for plots with sliding window size of 3 and 10. Again, activation patching is applied to the MLP activations at the last subject token. We find again that GN yields significantly more pronounced peak.

### H.2 PLOTS ON MLP PATCHING AT ALL TOKEN POSITIONS IN GPT-2 XL

See Figure 15–Figure 19 to plots on MLP patching at all token positions in GPT-2 XL, across window sizes of $3, 5, 10$. We observe that the right-side plots, using probability as the metric, highlights the last subject token as important. In contrast, the left-side figure using logit different does it to lesser degree.

### H.3 PLOTS ON SLIDING WINDOW PATCHING IN GPT2-XL

We provide further plots from our experiment that compares the sliding window patching with individual patching aggregated via summation over windows. See Figure 20–Figure 23.

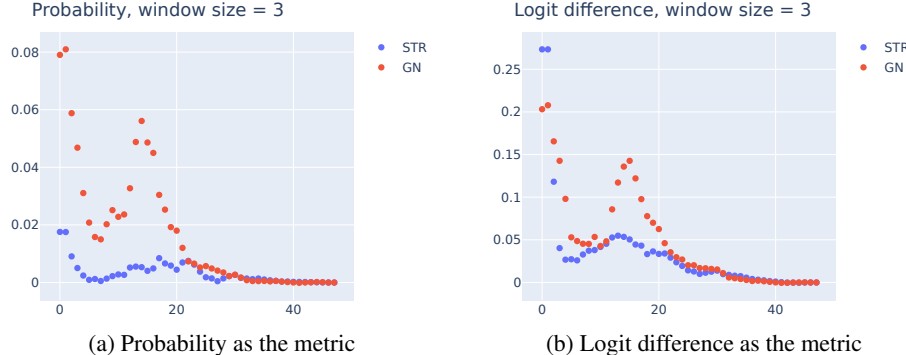

(a) Probability as the metric         (b) Logit difference as the metric

Figure 13: **MLP patching effects at the last subject token position** in GPT-2 XL on factual recall prompts, with window size of 3.

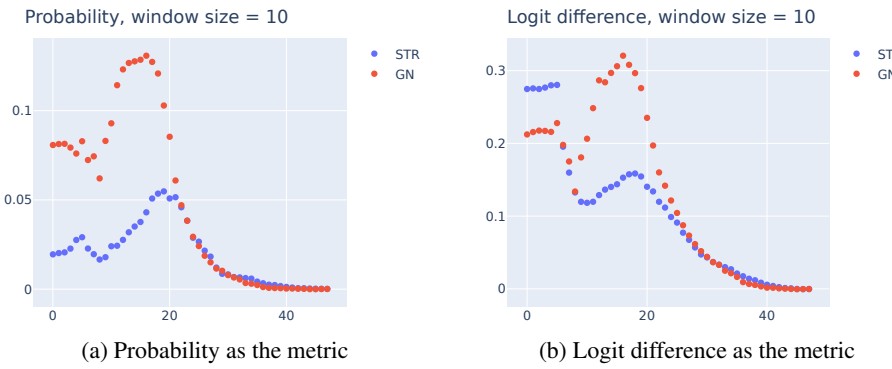

(a) Probability as the metric         (b) Logit difference as the metric

Figure 14: **MLP patching effects for factual recall** at the last subject token position in GPT-2 XL on factual recall prompts, with window size of 10.

## H.4   PLOTS ON ACTIVATION PATCHING OF MLP LAYERS ON GPT-2 LARGE

We perform activation patching on MLP layers of GPT-2 large in the factual association setting. Following our experiments in Section 3.1, we focus the effects at patching the MLP activation of the last subject token. We validate the high-level finding of Section 3.1, where we observe the disparity of GN and STR applied to MLP activation in the factual prediction setting. In particular, GN gives more pronounced concentration at early-middle MLP layers. We apply sliding window patching of size 3 and 5; see Figure 24 and Figure 25 for the resulting plots.

## H.5   PLOTS ON ACTIVATION PATCHING OF MLP LAYERS IN GPT-J

We perform activation patching on MLP layers of GPT-J (Wang & Komatsuzaki, 2021) in the factual association setting. We patch the MLP activations across all token positions and verify that probability tends to highlight the importance of the last subject token than logit difference. We focus on a sliding window patching of size 5 and the plots are given in Figure 27 (GN) and Figure 26 (STR). This complements our results in Section 4.

## I   FURTHER DETAILS ON IOI CIRCUIT DISCOVERY

### I.1   DETAILED PLOTS ON ACTIVATION PATCHING

We now provide the detailed plots from the activation patching experiments on the IOI circuit discovery task (Wang et al., 2023); see Figure 28 and Figure 29.

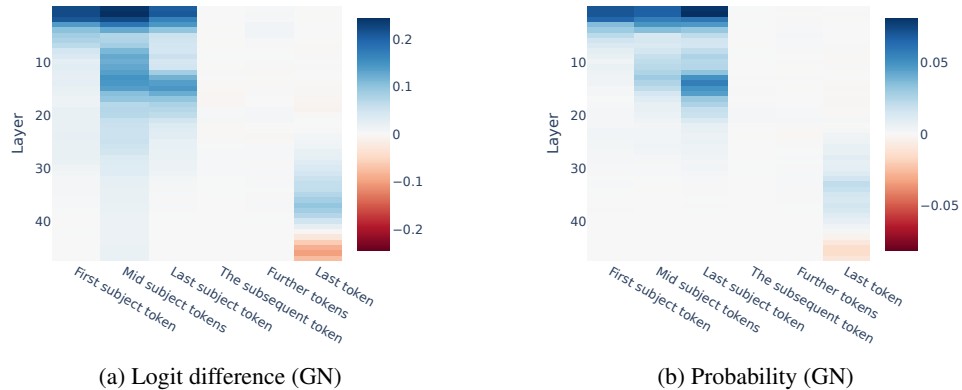

(a) Logit difference (GN)                    (b) Probability (GN)

Figure 15: **Activation patching on MLP** across layers and token positions in GPT-2 XL on factual recall prompts. Apply GN corruption and a sliding window of size 3.

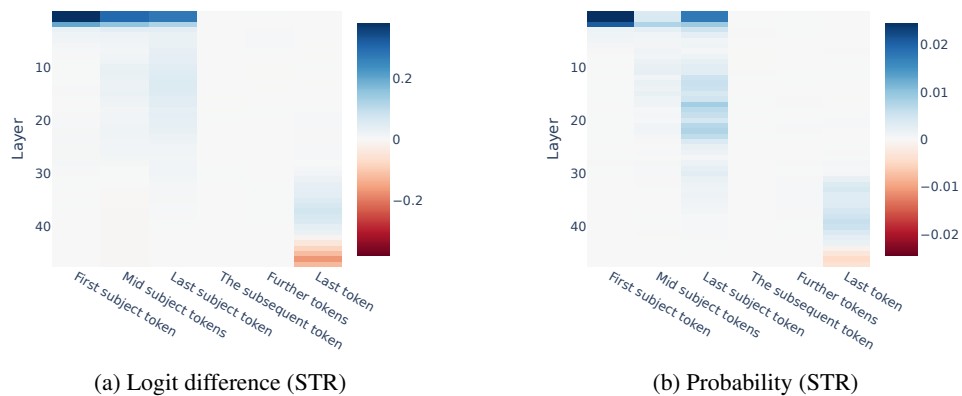

(a) Logit difference (STR)                   (b) Probability (STR)

Figure 16: **Activation patching on MLP** across layers and token positions in GPT-2 XL on factual recall prompts. Apply STR corruption and a sliding window of size 3.

## I.2 DETAILS ON DETECTIONS

We provide a detailed list of detection from attention heads patching in the IOI circuit setting (Section 3); see Table 5.

| Corruption | Metric | Negative heads | Positive heads |
|---|---|---|---|
| STR | Logit difference | 10.7, 11.10 | 5.5, 7.9, 8.6, 8.10, 9.9 |
| STR | Probability | 10.7 | 5.5, 7.9, 8.6, 8.10, 9.9 |
| STR | KL divergence | 10.7, 11.10 | 5.5, 7.9, 8.6, 8.10, 9.9 |
| GN | Logit difference | 10.7, 11.10 | 3.0, 5.5, 7.9, 8.6, 8.10, 9.9 |
| GN | Probability | 0.10, 10.7, 11.10 | 3.0, 5.5, 7.9, 8.6 |
| GN | KL divergence | 0.10, 10.7, 11.10 | 5.5, 7.9, 8.6 |

Table 5: **Detailed results from attention heads patching** in GPT-2 small on IOI sentences. A head is detected if the patching effect is two standard deviation from the mean effect. Negative heads are heads with negative patching effects, suggesting they hurt model performance.

## I.3 DETAILED PLOTS ON FULLY RANDOM CORRUPTION

We provide the plots on fully random corruption, termed $p_{ABC}$ in Wang et al. (2023). We perform activation patching on all attention heads, using both probability and logit difference as the metric

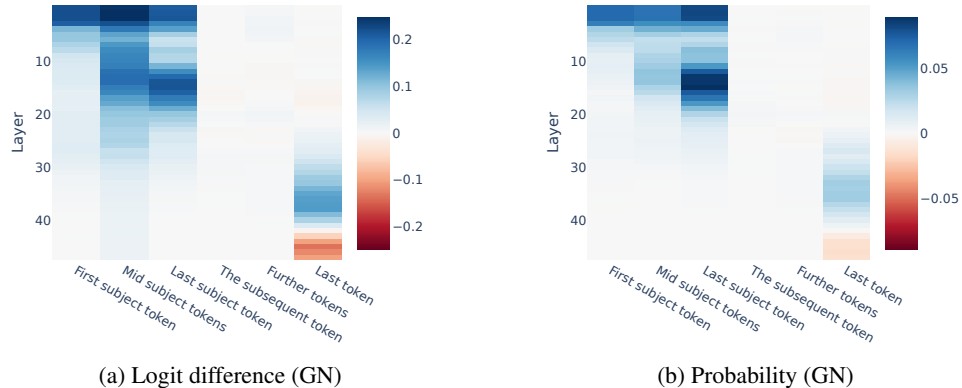

(a) Logit difference (GN)          (b) Probability (GN)

Figure 17: **Activation patching on MLP** across layers and token positions in GPT-2 XL on factual recall prompts. Apply GN corruption and a sliding window of size 5.

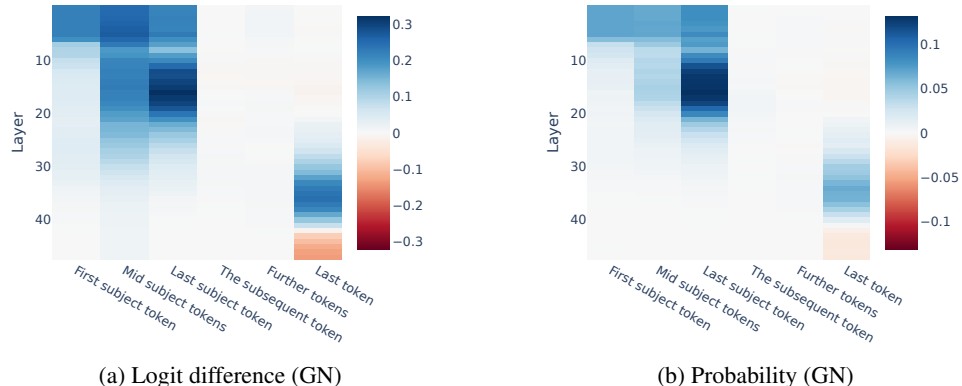

(a) Logit difference (GN)          (b) Probability (GN)

Figure 18: **Activation patching on MLP** across layers and token positions in GPT-2 XL on factual recall prompts. Apply GN corruption and a sliding window of size 10.

in order to draw contrasts between them. See Figure 30. In particular, we notice that there is no negative head in the plot. This is natural and totally expected, as we explained in Section 4.

## J  DATASET SAMPLES

**Factual data** We list a few dataset examples from the PAIREDFACTS dataset used in the factual recall experiments in Figure 31.[6] All the prompts are known true facts.

**IOI circuit** The detailed templates of constructing the $p_{\text{IOI}}$ data distribution can be found in Appendix A of Wang et al. (2023). We perform the same procedure of generating the IOI data by simply reusing their original code.

---

[6]The full dataset is available at https://www.jsonkeeper.com/b/P1GL.

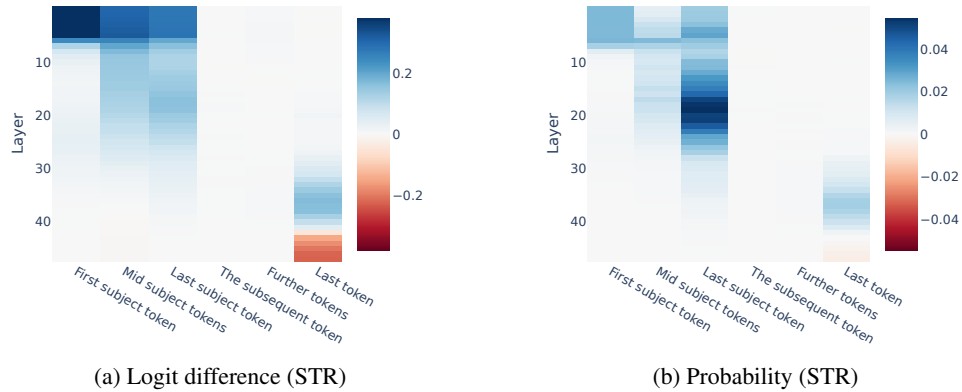

(a) Logit difference (STR)      (b) Probability (STR)

Figure 19: **Activation patching on MLP** across layers and token positions in GPT-2 XL. Apply STR corruption and a sliding window of size 10.

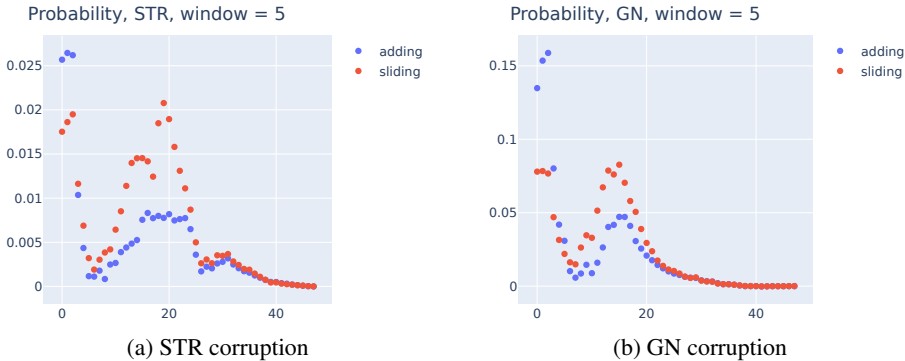

(a) STR corruption      (b) GN corruption

Figure 20: **MLP patching effects, sliding window vs summing up single-layer patching** at last token position in GPT-2 XL on factual recall prompts, with window size of 5. Apply probability as the metric.

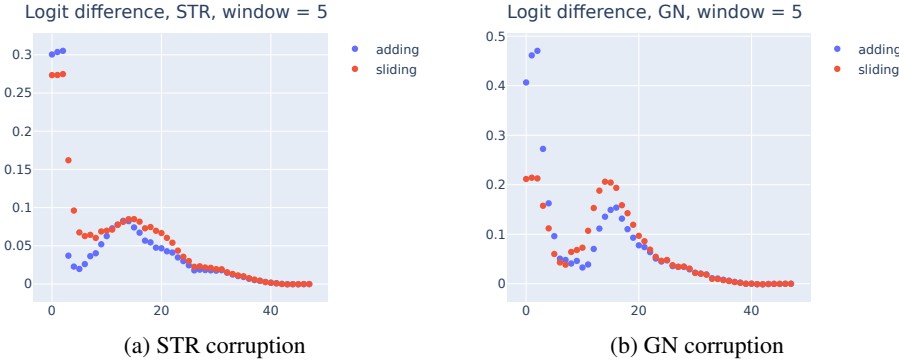

(a) STR corruption      (b) GN corruption

Figure 21: **MLP patching effects, sliding window vs summing up single-layer patching** at last token position in GPT-2 XL on factual recall prompts, with window size of 5. Apply logit difference as the metric.

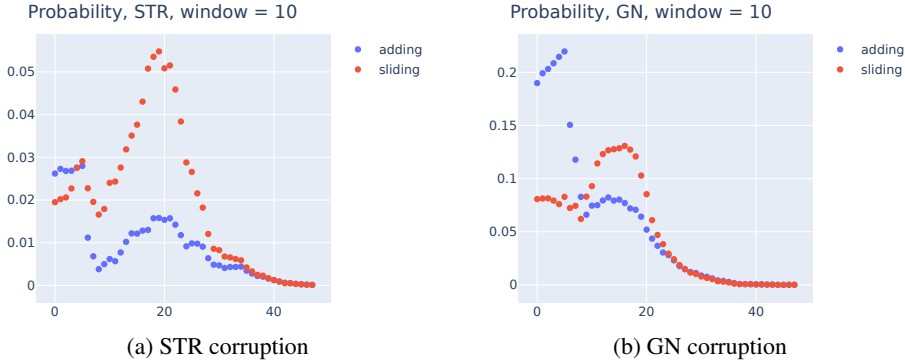

(a) STR corruption         (b) GN corruption

Figure 22: **MLP patching effects, sliding window vs summing up single-layer patching** at last token position in GPT-2 XL on factual recall prompts, with window size of 10. Apply probability as the metric.

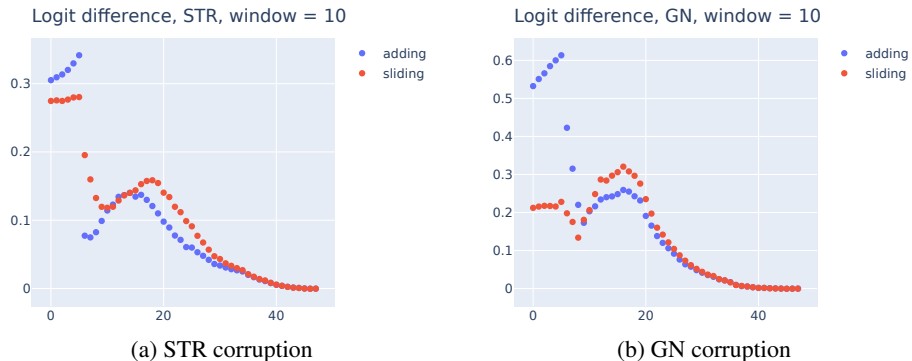

(a) STR corruption         (b) GN corruption

Figure 23: **MLP patching effects, sliding window vs summing up single-layer patching** at last token position in GPT-2 XL on factual recall prompts, with window size of 10. Apply logit difference as the metric.

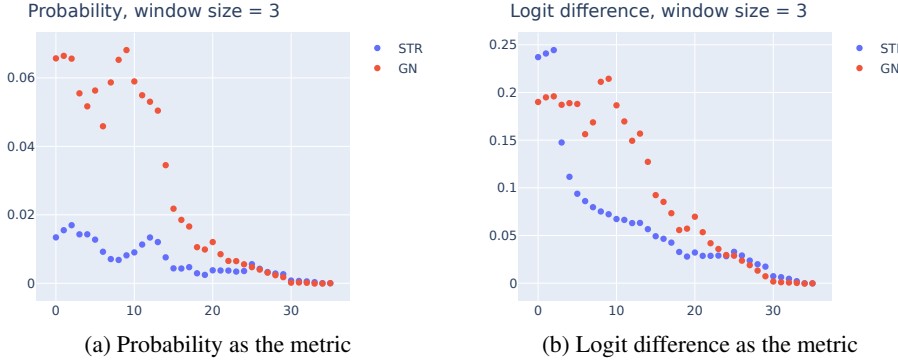

(a) Probability as the metric         (b) Logit difference as the metric

Figure 24: **MLP patching effects at the last subject token position** in GPT-2 large on factual recall prompts, with window size of 3.

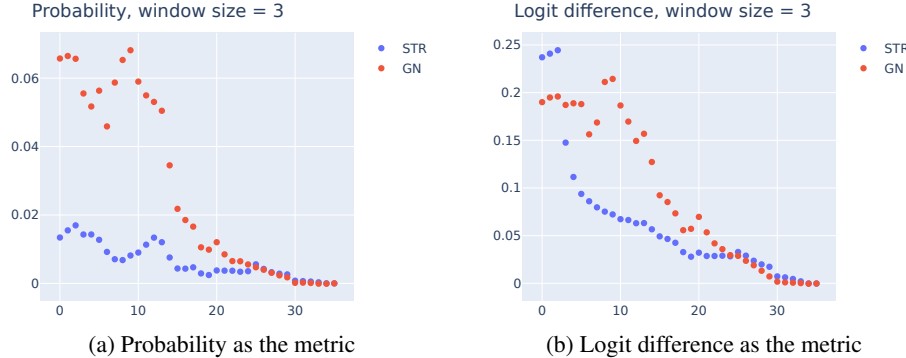

(a) Probability as the metric

(b) Logit difference as the metric

Figure 25: **MLP patching effects at the last subject token position** in GPT-2 large on factual recall prompts, with window size of 5.

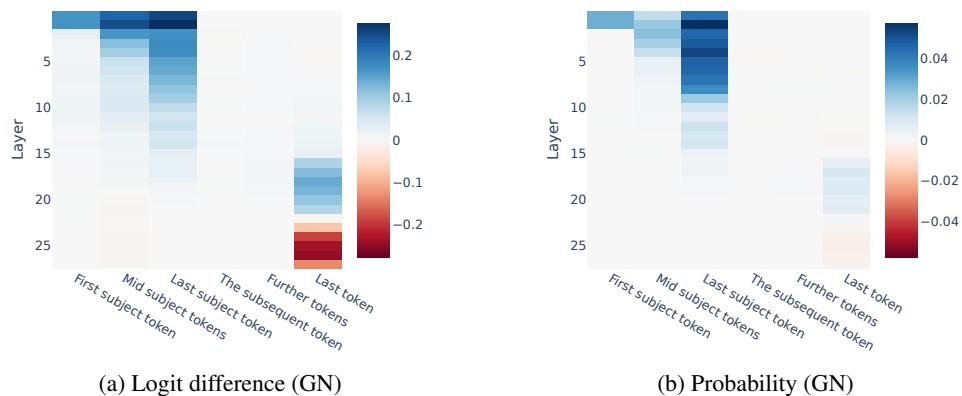

(a) Logit difference (GN)

(b) Probability (GN)

Figure 26: **Activation patching on MLP across layers and token positions in GPT-J** on factual recall prompts. Apply STR corruption and a sliding window of size 5.

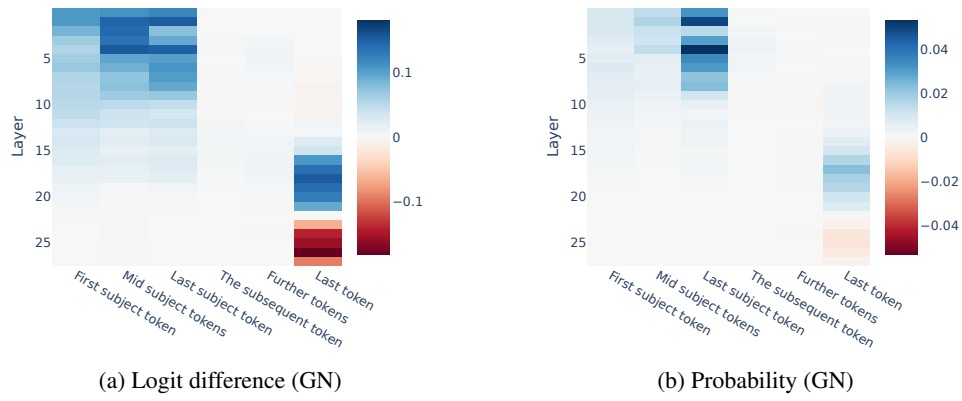

(a) Logit difference (GN)

(b) Probability (GN)

Figure 27: **Activation patching on MLP across layers and token positions in GPT-J** on factual recall prompts. Apply GN corruption and a sliding window of size 5.

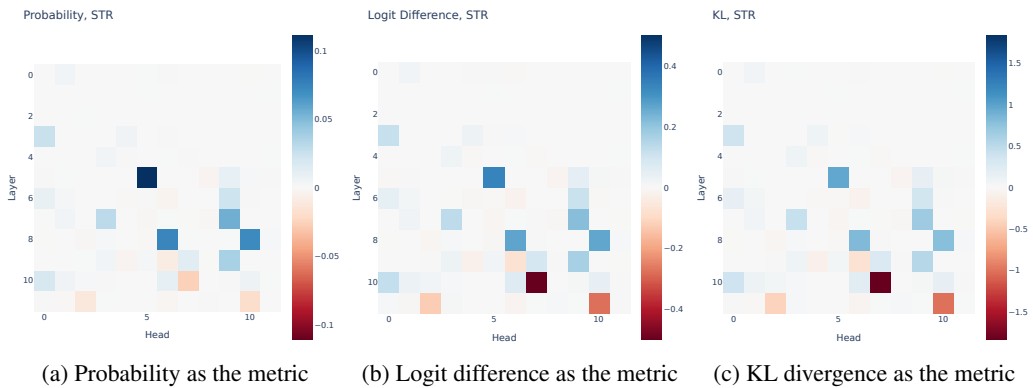

(a) Probability as the metric    (b) Logit difference as the metric    (c) KL divergence as the metric

Figure 28: **The effects of patching attention heads** in GPT-2 small using STR corruption on IOI sentences.

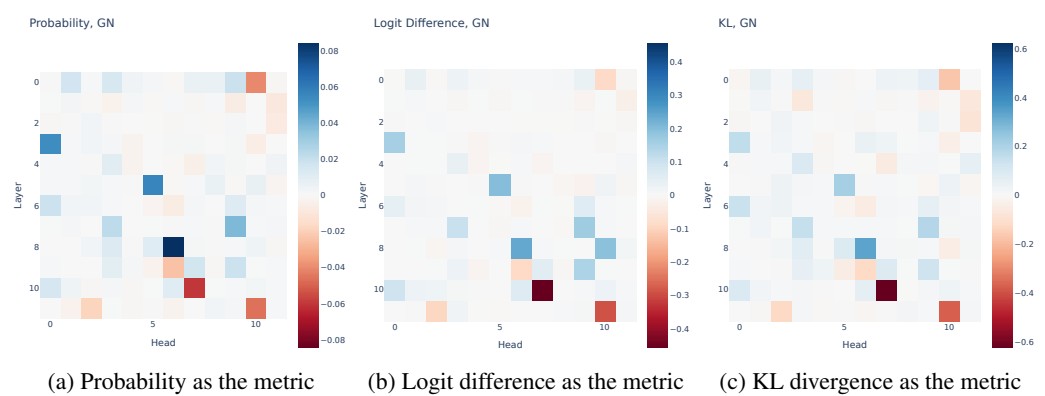

(a) Probability as the metric    (b) Logit difference as the metric    (c) KL divergence as the metric

Figure 29: **The effects of patching attention heads** in GPT-2 small using GN corruption on IOI sentences.

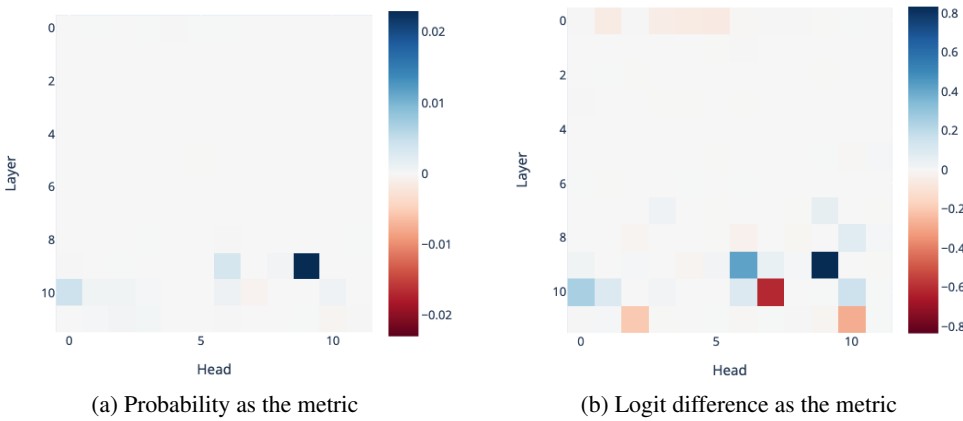

(a) Probability as the metric    (b) Logit difference as the metric

Figure 30: **The effects of patching attention heads** in GPT-2 small using fully random corruption on IOI sentences, with S1, S2 and IO replaced by three random names (denoted by $p_{ABC}$ in Wang et al. (2023)).

```
{
  "pair": [
    "Honus Wagner professionally plays the sport of",
    "Don Shula professionally plays the sport of"
  ],
  "answer": [
    " baseball",
    " football"
  ],
  "length": 9,
  "category": "athletes"
}

{
  "pair": [
    "Schreckhorn belongs to the continent of",
    "Afghanistan belongs to the continent of"
  ],
  "answer": [
    " Europe",
    " Asia"
  ],
  "length": 9,
  "category": "continents"
}
```

```
{
  "pair": [
    "Wii MotionPlus is developed by",
    "Chromebook Pixel is developed by"
  ],
  "answer": [
    " Nintendo",
    " Google"
  ],
  "length": 8,
  "category": "developers"
}

{
  "pair": [
    "The Eiffel Tower is in the city of",
    "Kinkakuji Temple is in the city of"
  ],
  "answer": [
    " Paris",
    " Kyoto"
  ],
  "category": "city_landmarks",
  "length": 11
}
```

Figure 31: **Sample text prompts** from the PAIREDFACTS dataset. The `length` field refers to the sequence length of the prompt under GPT-2 tokenizer.

