# OpenReview forum: "Towards Best Practices of Activation Patching in Language Models: Metrics and Methods"
_ICLR.cc/2024/Conference — ICLR 2024 poster_

### Official Review · Reviewer_PYwb · 2023-10-31

**Soundness:** 3 good
**Presentation:** 3 good
**Contribution:** 2 fair
**Rating:** 6
**Confidence:** 4

**Summary:**

In this paper the authors study the effects of different activation patching techniques on the mechanistic model interpretability and make several recommendations. They study and compare the selection of the hyperparameters, evaluation metrics and input corruption techniques for activation patching based model interpretability. The paper studies two approaches, Gaussian Noise (GN) and Semantic Token Replacement (STR), for corrupting the inputs. Based on those studies the paper recommends using STR input corruption technique since it produces in-distribution samples as opposed to the GN approach which produces OOD samples. In terms of the evaluation metrics, it recommends logit difference since it is more granular and allows to detect model activations (components) that have negative impact on model performance.

**Strengths:**

1) The abstract and introduction are well and clearly written. The problem statement and contributions are easy to follow from those 2 sections.
2) The paper performs thorough experimentation on sliding window techniques, localizing factual recall and circuit discovery.

**Weaknesses:**

1) The work overall is very interesting but it feels a bit light on the novelty. Perhaps proposing additional novel methods for input corruption and improvements for the activation patching techniques can help to increase the novelty in this paper.
2) The way the paper is written it might be a good fit for a workshop.
3) Some terminology could be explained in the paper. E.g. Name Mover
It seems that the paper requires prior knowledge of another paper Wang et.al. Some concepts such as: 0.10 negative detection is not very clear.
4) It might be good to describe clearly what 0.10 negative detection is under the `Negative detection of 0.10 under GN` section.



Minor comments:
1) “use its own the method of generating” -> “use its own method of generating” ?

**Questions:**

1) What are some of the novel contributions of the paper ?
2) How was the content of Table 1 computed ?

---

> ### Author Response · Authors · 2023-11-11
>
> Thank you for reviewing our work! We address your concerns below.
>
> > The work overall is very interesting but it feels a bit light on the novelty. Perhaps proposing additional novel methods for input corruption and improvements for the activation patching techniques can help to increase the novelty in this paper.
>
>
> We believe that our work is significant, since it gives the first rigorous and comprehensive evaluation of a foundational technique.  As we surveyed in the paper and [a rebuttal comment above](https://openreview.net/forum?id=Hf17y6u9BC&noteId=swlVw4U9Z4), a large number of recent papers in interpretability have all used activation patching (AP).
> Yet, many works use somewhat different variants in terms of metric or implementation details. Prior to our work, there was no comprehensive study on the consistency of AP. Our paper fills this gap by (i) rigorously evaluating a wide range of techniques within AP and (ii) providing recommendations for its best practices. Hence, while the main contribution of our work may be unglamorous, we believe it is crucial as it provides the rigor for the field moving forward.
>
> In terms of novelty, our investigation into GN corruption suggests that it may introduce OOD inputs to the model's internal components. We further argue that it may lead to unreliable interpretability claims. To the best of our knowledge, this insight is novel in the literature. It provides an importance evidence for the limitation of the technique.
>
> > The way the paper is written it might be a good fit for a workshop.
>
> Our hope was to write a paper that would be of value to practitioners in LM interpretability who use activation patching in their research, as well as of broad interest to the community. We would appreciate any advice on improving the presentation of our paper.
>
> > Some terminology could be explained in the paper. E.g. Name Mover It seems that the paper requires prior knowledge of another paper Wang et.al. Some concepts such as: 0.10 negative detection is not very clear.
>
> > It might be good to describe clearly what 0.10 negative detection is under the Negative detection of 0.10 under GN section.
>
> Thanks for pointing this out! In the end of the first paragraph of Section 3.2, we added the definition that “we say a detection is negative if the patching effect of the component is negative (under a given metric).” Intuitively, this means that patching suggests the component hurts model performance. We also added Appendix B, an overview of the attention heads in the IOI circuit of GPT-2 small, as found by Wang et al. We hope these clarify the setting of Table 1.
>
> > How was the content of Table 1 computed?
>
> As detailed in Section 3.1, we say that a head is detected if its patching effect is 2 standard deviations (SD) away from the mean effect. We have also clarified this in the main text: Table 1 is computed by patching each individual head, calculating the patching effects, listing the heads that are detected (according to the definition above) and contrasting with the results from Wang et al.

---

> > ### Comment · Reviewer_PYwb · 2023-11-21
> > **Response to the authors**
> >
> > Thank you very much for addressing my comments.
> >
> > 1. For Table 1. it would be good to explain what stands on the left and right sides of ' / '. At this point it's a bit hard to understand.
> > 2. In a general case, does the paper propose what's the best way of choosing Symmetric token replacement (STR). This could be a time consuming and manual task for many use cases, wouldn't it ?

---

> > > ### Author Response · Authors · 2023-11-21
> > >
> > > Thank you for engaging with the discussion!
> > >
> > > 1. In Table 1, the numerator is the the number of detections from our method, and the denominator is the findings from Wang et al (which is more exhaustive than a single round of activation patching that we attempt here). For example, 1/3 NM indicates 1 out of 3 NMs is detected. We have updated the draft to clarify this point.
> > >
> > > 2. In a general case, we make the suggestion that STR should attempt at different token positions if possible. Constructing paired prompts for STR is not always difficult. If the task is synthetic, such as IOI or greater-than, it's straightforward to generate data for STR, by swapping the key information token with a similar one from the sample distribution. More generally, one can first construct a large database of single prompts and pair them up based on token alignment in an automated way. In fact, this is how we produced PairedFacts from the original datasets of [Meng et al](https://rome.baulab.info/). One can also leverage LLM, though we did not explore this option in this work.
> > >
> > > Please let us know if you have further questions! At this stage, we are still able to make changes to the draft and actively participate in discussion.

---

> > > > ### Comment · Reviewer_PYwb · 2023-11-21
> > > > **Response to authors**
> > > >
> > > > Thank you very much for addressing my comments.
> > > > I increased my score by one point.

---

### Official Review · Reviewer_bjM9 · 2023-10-31

**Soundness:** 4 excellent
**Presentation:** 3 good
**Contribution:** 3 good
**Rating:** 8
**Confidence:** 3

**Summary:**

In this work, the authors explore how the method of corruption and type of evaluation can cause conflicting take-aways from activation patching. Specifically, they look at two ways to add corruption to a token embedding: gaussian noise (adding noise to the embedding) or symmetric token replacement (replacing a token with a semantically related token). They find that gaussian noise can cause the input to be OOD, thus breaking the internal mechanism. When looking at logit difference vs. probability, they find that probability can overlook negative model components. Finally they look at sliding window patching, and find that it can inflate logit plots.

**Strengths:**

I like the motivation of this paper: I think its important to understand how hyperparameters can change the results of interpretability methods.

The authors go over several different types of hyperparameter (corruption method, evaluation method, sliding window) and convincingly show that these design choices can result in different interpretations. I liked the Name Mover analysis, which made it easier to understand how gaussian noise could be causing issues for activation patching.

Finally, I appreciated Section 6, which gives recommendations on how activation patching should be performed.

**Weaknesses:**

My biggest concern is relevance to the community: other than Meng et. al, are there other papers using gaussian noise? It's not clear to me that this is a wide-spread issue.

Moreover, the recommended course of action (STR) can be more difficult to actually implement (as it requires having semantically similar substitutions). To be of most relevance to the community, it would be great if the authors suggested an approach that had the flexibility of gaussian noise without introducing as much bias.

Clarity: Table 1 and in particular the part about negative detection was a bit hard to parse. I would make the discussion around those results more clear.

**Questions:**

How much does different substitutions for STR or different samples of noise (for gaussian noise) change the interpretation? Are the methods at least consistent within themselves?

---

> ### Author Response · Authors · 2023-11-11
>
> Thank you for reading and appreciating our work! We address your questions and concerns below.
> > My biggest concern is relevance to the community: other than Meng et. al, are there other papers using gaussian noise? It's not clear to me that this is a wide-spread issue.
>
> We remark that the follow-up work of [Meng et al (2023)](https://arxiv.org/abs/2210.07229) and [Hase et al (2023)](https://arxiv.org/abs/2301.04213) both use activation patching with Gaussian noise.
>
> In fact, [Meng et at (2022)](https://rome.baulab.info/), though fairly recent, is quite influential in the community, accumulating nearly 200 citations over the year. Thus, we believe more follow-ups of it may emerge in the future.
>
> > Moreover, the recommended course of action (STR) can be more difficult to actually implement.
>
> We agree, but we also remark that this really depends on specific tasks. In the case of IOI, STR is not difficult to implement, as demonstrated by Wang et al. One can simply identify a collection of single-token names and use them for corruption. In the case of factual recall, it takes some effort, but in our work, we have managed to construct a dataset of 145 symmetric pairs of prompts, as detailed in Appendix C.
>
> > To be of most relevance to the community, it would be great if the authors suggested an approach that had the flexibility of gaussian noise without introducing as much bias.
>
> One possibility is mean ablation and resample ablation. Here, mean ablation replaces the activation of a model component with the average activation (over a dataset), and
> resample activation averages the effects of different ablations; see [Chan et al 2022](https://www.alignmentforum.org/posts/JvZhhzycHu2Yd57RN/causal-scrubbing-a-method-for-rigorously-testing) for further discussions.
>
>  We have added a note about this in the paper. We leave it as a future direction to compare ablations and activation patching for circuit analysis.
>
> > Clarity: Table 1 and in particular the part about negative detection was a bit hard to parse. I would make the discussion around those results more clear.
>
> Thanks for pointing this out! In the end of the first paragraph of Section 3.2, we added the definition that “we say a detection is negative if the patching effect of the component is negative (under a given metric).” Intuitively, this means that patching suggests the component hurts model performance. We also added Appendix B, an overview of the attention heads in the IOI circuit of GPT-2 small, as found by Wang et al. We hope these clarify the setting of Table 1.
>
> > How much does different substitutions for STR or different samples of noise (for gaussian noise) change the interpretation? Are the methods at least consistent within themselves?
>
> This is a great question! In section 6, we investigate the effect of varying the corrupted token position in STR. For example, instead of corrupting S2 (as done in section 3), what happens if we corrupt S1 and IO? As shown in Appendix G, the experiments suggest this leads to different attention heads discoveries. Moreover, in Appendix I.3, we experiment with corrupting all S1, S2 and IO by three random names, in the same way as Wang et al. The results are again somewhat different. We believe, however, that all these discoveries are different perspectives on the same circuit and simply complement each other, since different corruptions just trace different information; see Section 6 for a conceptual argument of the point.
>
> For GN, [the original work of Meng et al](https://rome.baulab.info/) attempt at different noise levels and noise distributions and find pretty consistent results. We reached the same conclusion in our experiments, and hence didn’t delve into this issue again in the paper.

---

> > ### Comment · Reviewer_bjM9 · 2023-11-21
> > **Response**
> >
> > Thank you for your response. The authors have answered my questions. I have raised my score to an accept (8).

---

### Official Review · Reviewer_4A2t · 2023-11-03

**Soundness:** 3 good
**Presentation:** 3 good
**Contribution:** 2 fair
**Rating:** 6
**Confidence:** 2

**Summary:**

This paper delves into the realm of mechanistic interpretability, a burgeoning and promising domain within large language models. It primarily centers on activation patching, aiming to identify activations that hold a causal influence over the output. A notable aspect of this work is its pioneering stance in systematically studying the generation of corrupted prompts and the evaluation metrics for patching effects, which previously lacked standardization. Specifically, the paper scrutinizes two methodologies for generating corrupted prompts: 1) Gaussian Noising (GN) and 2) Symmetric Token Replacement (STR). Furthermore, it explores two evaluation metrics: 1) probability and 2) logit difference, alongside investigating the impact of sliding window patching.

**Strengths:**

The endeavor to understand the internal mechanisms of large language models through activation patching is pivotal. This paper stands out by empirically examining various methodologies, bridging the gap where variations across different papers have made it challenging to ascertain the more effective approach. By embarking on this comprehensive investigation, the paper makes a substantial contribution towards standardizing methods, which is invaluable to the mechanistic interpretability community.

**Weaknesses:**

I am not very familiar with the details of the existing activation patching methods. Therefore, I am not sure whether the methods included in the paper are diverse and representative enough for the mechanistic interpretability community.

**Questions:**

N/A

---

> ### Author Response · Authors · 2023-11-11
>
> Thank you for evaluating and appreciating our work!
>
> > I am not very familiar with the details of the existing activation patching methods. Therefore, I am not sure whether the methods included in the paper are diverse and representative enough for the mechanistic interpretability community.
>
> First, we note that activation patching (AP), as a generic method, is foundational and widely used in mechanistic interpretability. In particular, the third paragraph of our introduction lists 5 prior works that apply the method. We also refer the reviewer to consider the related work section that gives more examples. In total, one can  identify over 20 papers using AP or its variants in the literature (which [we list in a separate comment below](https://openreview.net/forum?id=Hf17y6u9BC&noteId=swlVw4U9Z4)).
>
> Second, the metrics and particular variants considered by our work cover a large fraction of existing techniques in AP. Specifically, this includes techniques used in [Meng et al., 2022](https://rome.baulab.info/), [Wang et al., 2023](https://arxiv.org/abs/2211.00593), [Hanna et al., 2023](https://arxiv.org/abs/2305.00586), [Heimersheim & Janiak, 2023](https://www.alignmentforum.org/posts/u6KXXmKFbXfWzoAXn/a-circuit-for-python-docstrings-in-a-4-layer-attention-only) and [Stolfo et al., 2023](https://arxiv.org/abs/2305.15054). All of them are studied in detail in our paper. As we surveyed in the related work section and the comment below, these same techniques are used in other works as well. In addition, we also investigate KL divergence as a metric, used in [Conmy et al., 2023](https://arxiv.org/abs/2304.14997).
>
> Finally, in section 6, we consider the effect of varying the corrupted token position, a rather neglected aspect in the literature. This again adds to the comprehensiveness of our study.

---

> > ### Author Response · Authors · 2023-11-11
> > **A list of papers that use activation patching**
> >
> > 1. Jesse Vig, Sebastian Gehrmann, Yonatan Belinkov, Sharon Qian, Daniel Nevo, Yaron Singer, and Stuart Shieber. Investigating gender bias in language models using causal mediation analysis. In Advances in Neural Information Processing Systems (NeurIPS), 2020
> > 2. Atticus Geiger, Hanson Lu, Thomas Icard, and Christopher Potts.  Causal abstractions of neural  networks. In  Advances in Neural Information Processing Systems (NeurIPS), 2021
> > 3. Atticus Geiger, Kyle Richardson, and Christopher Potts. Neural natural language inference models partially embed theories of lexical entailment and negation. In Proceedings of the Third BlackboxNLP Workshop on Analyzing and Interpreting Neural Networks for NLP, 2020
> > 3. Paul Soulos, R Thomas McCoy, Tal Linzen, and Paul Smolensky. Discovering the compositional  structure of vector representations with role learning networks. In  Proceedings of the Third BlackboxNLP Workshop on Analyzing and Interpreting Neural Networks for NLP, 2020
> > 4. Matthew Finlayson, Aaron Mueller, Sebastian Gehrmann, Stuart M Shieber, Tal Linzen, and Yonatan Belinkov. Causal analysis of syntactic agreement mechanisms in neural language models. In Proceedings of the 59th Annual Meeting of the Association for Computational Linguistics and the 11th International Joint Conference on Natural Language Processing (ACL-IJCNLP), 2021.
> > 5. Atticus Geiger, Zhengxuan Wu, Hanson Lu, Josh Rozner, Elisa Kreiss, Thomas Icard, Noah Goodman, and Christopher Potts. Inducing causal structure for interpretable neural networks. In International Conference on Machine Learning (ICML), 2022.
> > 6. Kevin Meng, David Bau, Alex Andonian, and Yonatan Belinkov. Locating and editing factual associations in GPT. In Advances in Neural Information Processing Systems (NeurIPS), 2022
> > 7. Kevin Ro Wang, Alexandre Variengien, Arthur Conmy, Buck Shlegeris, and Jacob Steinhardt. Interpretability in the wild: a circuit for indirect object identification in GPT-2 small. In  International  Conference on Learning Representations (ICLR), 2023
> > 8. Peter Hase, Mohit Bansal, Been Kim, and Asma Ghandeharioun. Does localization inform editing?  Surprising differences in causality-based localization vs. knowledge editing in language models.  In  Advances in Neural Information Processing Systems (NeurIPS), 2023
> > 9. Michael Hanna, Ollie Liu, and Alexandre Variengien.  How does GPT-2 compute greater-than?:  Interpreting mathematical abilities in a pre-trained language model. In  Advances in Neural Information Processing Systems (NeurIPS), 2023
> > 10. Arthur Conmy, Augustine N Mavor-Parker, Aengus Lynch, Stefan Heimersheim, and Adri`a Garriga-Alonso.  Towards automated circuit discovery for mechanistic interpretability.  In  Advances in  Neural Information Processing Systems (NeurIPS), 2023
> > 10. Todd, Eric, Millicent L. Li, Arnab Sen Sharma, Aaron Mueller, Byron C. Wallace, and David Bau. "Function Vectors in Large Language Models." arXiv preprint arXiv:2310.15213 (2023).
> > 11. Hendel, Roee, Mor Geva, and Amir Globerson. "In-Context Learning Creates Task Vectors." arXiv preprint arXiv:2310.15916 (2023).
> > 11. Feng, Jiahai, and Jacob Steinhardt. "How do Language Models Bind Entities in Context?." arXiv preprint arXiv:2310.17191 (2023).
> > 12. Alessandro Stolfo, Yonatan Belinkov, and Mrinmaya Sachan. Understanding arithmetic reasoning  in language models using causal mediation analysis.  arXiv preprint arXiv:2305.15054, 2023
> > 13. Hoagy Cunningham, Aidan Ewart, Logan Riggs, Robert Huben, and Lee Sharkey. Sparse autoencoders find highly interpretable features in language models. arXiv preprint arXiv:2309.08600, 2023
> > 14. Tom Lieberum, Matthew Rahtz, Janos Kramar, Geoffrey Irving, Rohin Shah, and Vladimir Mikulik.  Does circuit analysis interpretability scale? Evidence from multiple choice capabilities in  Chinchilla.  arXiv preprint arXiv:2307.09458, 2023
> > 15. Nicholas Goldowsky-Dill, Chris MacLeod, Lucas Sato, and Aryaman Arora. Localizing model behavior with path patching. arXiv preprint arXiv:2304.05969, 2023
> > 16. Mor Geva, Jasmijn Bastings, Katja Filippova, and Amir Globerson. Dissecting recall of factual associations in auto-regressive language models. arXiv preprint arXiv:2304.14767, 2023
> > 17. Huang, Jing, Atticus Geiger, Karel D'Oosterlinck, Zhengxuan Wu, and Christopher Potts. Rigorously Assessing Natural Language Explanations of Neurons. arXiv preprint arXiv:2309.10312 (2023).
> > 18. Merullo, Jack, Carsten Eickhoff, and Ellie Pavlick. "Circuit Component Reuse Across Tasks in Transformer Language Models." arXiv preprint arXiv:2310.08744 (2023).
> > 19. Tigges, Curt, Oskar John Hollinsworth, Atticus Geiger, and Neel Nanda. "Linear Representations of Sentiment in Large Language Models." arXiv preprint arXiv:2310.15154 (2023).
> > 20. McDougall, Callum, Arthur Conmy, Cody Rushing, Thomas McGrath, and Neel Nanda. "Copy Suppression: Comprehensively Understanding an Attention Head." arXiv preprint arXiv:2310.04625 (2023).

---

### Author Response · Authors · 2023-11-11

We thank the reviewers for their thoughtful comments and valuable feedback. We have responded to the comments and made changes to our submission. In particular:

* We have clarified the setting of Section 3.2 by (i) adding a definition of "negative detection" and (ii) adding Appendix B which provides a quick overview of the attention heads in the IOI circuit.
* Reviewer PYwb raised concerns about the novelty of our study. We clarify that, to our knowledge, our paper is the first rigorous and comprehensive evaluation of the foundational method of activation patching (AP). We consider this as  an important step, as [many recent works have applied this method](https://openreview.net/forum?id=Hf17y6u9BC&noteId=swlVw4U9Z4), and yet standardization seems lacking.


We also note that we appreciate the positive remarks, such as
* The paper is a comprehensive investigation and makes a substantial contribution towards standardizing methods. (Reviewer 4A2t)
* The research topic is well motivated. (Reviewer bjM9)
* The main results on different interpretability outcomes are convincing. (Reviewer bjM9)
* The Name Mover analysis made it easier to understand how Gaussian noise could be causing issues for activation patching. (Reviewer bjM9)
* The paper provides recommendations on how activation patching should be performed. (Reviewer bjM9)
* The abstract and introduction are well written. (Reviewer PYwb)
* The experiments are thorough. (Reviewer PYwb)

We provide our responses to the initial comments of each reviewer below. We hope our answers resolve all initial questions and concerns raised by the reviewers and we will be most happy to answer any remaining questions during the discussion phase!

---

### Author Response · Authors · 2023-11-19
**Discussion period ending**

As the end of the discussion period is approaching (November 22, in 3 days), this is a gentle reminder to the reviewers to check out our rebuttals. If there are any questions or concerns, we are most happy to continue the discussion!

---

### Meta-Review · Area_Chair_TvB8 · 2023-12-05

**Metareview:**

The authors study mechanistic interpretability methods with the goal of understanding the impact of implementation choices such as hyperparameters and corruption methods. The findings indicate sensitivity to hyperparameters and suggests the use of STR perturbations with logit difference evals.

Strengths of the paper is the careful experiments on the impact of various design decisions.
Weaknesses are generally centered on the impact of the work - whether these hyperparameter issues are a real problem in practice and whether the analysis in the paper is 'enough' or if there should be some kind of methods component to the paper.

**Justification For Why Not Higher Score:**

There is a very real question of how much of the community has been / will be affected by these issues. As the authors note in the rebuttal, most of the work on gaussian noise perturbations is by Meng et al.

**Justification For Why Not Lower Score:**

The reviewers are generally fairly positive about the paper post-rebuttal.

---

### Decision · Program_Chairs · 2024-01-16

Accept (poster)